# Insect infestations and the persistence and functioning of oak-pine mixedwood forests in the mid-Atlantic region, USA

**Kenneth L. Clark**[1]*, **Carissa Aoki**[2], **Matthew Ayres**[3], **John Kabrick**[4], **Michael R. Gallagher**[1]

**1** USDA Forest Service, Northern Research Station, Silas Little Experimental Forest, New Lisbon, New Jersey, United States of America, **2** Environmental Studies, Bates College, Lewiston, Maine, United States of America, **3** Department of Biological Sciences, Dartmouth College, Hanover, New Hampshire, United States of America, **4** USDA Forest Service, Northern Research Station, Columbia, Missouri, United States of America

* kenneth.clark@usda.gov

**Data Availability Statement:** Most of the carbon and water flux data used in our analyses are already available at the Ameriflux data archive (https://ameriflux.lbl.gov/sites/siteinfo/US-Slt,

## Abstract

Damage from infestations of *Lymantria dispar* L. in oak-dominated stands and southern pine beetle (*Dendroctonus frontalis* Zimmermann) in pine-dominated stands have far exceeded impacts of other disturbances in forests of the mid-Atlantic Coastal Plain over the last two decades. We used forest census data collected in undisturbed and insect-impacted stands combined with eddy covariance measurements made pre- and post-disturbance in oak-, mixed and pine-dominated stands to quantify how these infestations altered forest composition, structure and carbon dynamics in the Pinelands National Reserve of southern New Jersey. In oak-dominated stands, multi-year defoliation during *L. dispar* infestations resulted in > 40% mortality of oak trees and the release of pine saplings and understory vegetation, while tree mortality was minimal in mixed and pine-dominated stands. In pine-dominated stands, southern pine beetle infestations resulted in > 85% mortality of pine trees but had minimal effect on oaks in upland stands or other hardwoods in lowland stands, and only rarely infested pines in hardwood-dominated stands. Because insect-driven disturbances are both delaying and accelerating succession in stands dominated by a single genus but having less effect in mixed-composition stands, long-term disturbance dynamics are favoring the formation and persistence of uneven age oak-pine mixedwood stands. Changes in forest composition may have little impact on forest productivity and evapotranspiration; although seasonal patterns differ, with highest daily rates of net ecosystem production (NEP) during the growing season occurring in an oak-dominated stand and lowest in a pine-dominated stand, integrated annual rates of NEP are similar among oak-, mixed and pine-dominated stands. Our research documents the formation of mixedwood stands as a consequence of insect infestations in the mid-Atlantic region and suggests that managing for mixedwood stands could reduce damage to forest products and provide greater continuity in ecosystem functioning.

https://ameriflux.lbl.gov/sites/siteinfo/US-Dix, and
https://ameriflux.lbl.gov/sites/siteinfo/US-Ced for
the oak, mixed and pine stands, respectively. Much
of the forest census data for the three sites
impacted by gypsy moth are available from the
USDA Forest Service Research Data Archive, Fort
Collins, CO: https://www.fs.usda.gov/rds/ archive/.

**Funding:** Partial support for this project was
provided by USDA Forest Service Forest Health and
Monitoring Program (https://www.fs.fed.us/
foresthealth/) grants NE-EM-F-13-01 to KC and NE-
EM-B-12-01 to MA and AK. The funders had no
role in study design, data collection and analysis,
decision to publish, or preparation of the
manuscript.

**Competing interests:** The authors have declared
that no competing interests exist.

# Introduction

Throughout the Northeast and mid-Atlantic regions of the USA, intermediate age forests with median tree ages of approximately 70 to 110 years have regenerated following farm abandonment, the cessation of industrial forestry practices such as clearcutting and charcoal production, and a decrease in the occurrence of severe wildfires [1–3]. Disturbance regimes in these forests differ in spatial and temporal scales, and intensity, compared to previous stand replacing disturbances, and are now characterized by insect infestations, disease, windstorms, managed wildland fire, and harvesting [4–7]. Impacts resulting from infestations of native and non-native insects are especially acute in the northeastern US, as they now account for the majority of forest damage [8–11]. On the mid-Atlantic Coastal Plain, oak (*Quercus* spp.) tree mortality resulting from infestations of *L. dispar* (*Lymantria dispar* L.) in oak-dominated stands and pine (*Pinus* spp.) mortality following infestations of southern pine beetle (*Dendroctonus frontalis* Zimmermann) in pine-dominated stands over the last decade have far exceeded the area impacted by wildfires, harvesting or windstorms, which were previously the major disturbances in these forests [12–15].

The 445,000 ha Pinelands National Reserve (PNR) in southern New Jersey contains the largest forested area on the mid-Atlantic Coastal Plain. Following the cessation of intensive forest harvesting and charcoaling activities, less frequent wildfires because of suppression activities and changes in forest management practices have facilitated the establishment and persistence of oaks and other mesic hardwoods in the PNR (Fig 1; [12, 16–18]). More recently, oak tree and sapling mortality in oak-dominated stands infested by *L. dispar* have facilitated the release and regeneration of pines, leading to the formation of uneven-age mixed composition stands. These "mixedwoods" are characterized by neither hardwood nor softwood species exceeding approximately 75% dominance [e.g., 19–22]. Numerous pine-dominated stands established naturally following harvesting, charcoaling, and then repeated severe wildfires early in the 20th century. Continued wildfire activity and landscape-scale prescribed burning has limited the regeneration of oaks and other mesic hardwoods and resulted in the persistence of pine-dominated stands [12, 16, 17, 23]. Over the last two decades, pine tree mortality as a result of southern pine beetle infestations in previously pine-dominated stands has increased the proportional basal area (BA; m$^{-2}$ ha$^{-1}$) and biomass of oaks in upland stands, and of hardwoods such as red maple (*Acer rubrum* L.) and black gum (*Nyssa sylvatica* Marshall) in lowland stands, also resulting in the formation of uneven-age mixedwood stands (Fig 1; [15, 24, 25]).

Insect infestations that target specific softwood or hardwood species have short- and long-term effects on the functioning of forest ecosystems [13, 26–30]. In the absence of infestations or other major disturbance, annual net primary productivity (NPP) of intermediate age forests in the mid-Atlantic region derived from USDA Forest Inventory and Analysis data (FIA; [9]) and FIA-type forest census plots in the PNR range from 3.8 to 4.6 T C ha$^{-1}$ yr$^{-1}$; estimates for oak-dominated, oak-pine mixedwood, and pine-dominated forests from both sources are similar (S1 Table). Simulated values of NPP for oak-dominated, oak-pine mixedwood, and pine-dominated stands using three different process-based models are consistent with forest census estimates of NPP [31–34]. Estimated net ecosystem production (NEP) by oak-dominated, oak-pine mixedwood, and pine-dominated stands across the region derived from FIA data and model simulations range from 1.2 to 2.3 T C ha$^{-1}$ yr$^{-1}$ (S1 Table). Derived and simulated NEP estimates are consistent with annual NEP values calculated from eddy covariance measurements of net ecosystem exchange of $CO_2$ (NEE) during undisturbed years in intermediate age oak-dominated, oak-pine mixedwood, and pine-dominated stands in the PNR [13, 30] (S1 Table).

Short-term impacts of insect infestations on ecosystem functioning of mid-Atlantic forests have been well-characterized using forest census, remote sensing, and carbon flux

## Conceptual model of disturbance and persistence of mixedwood stands in the Pinelands National Reserve, New Jersey

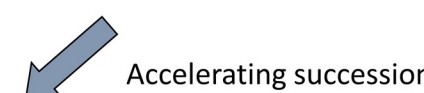

Low fire frequency/intensity

**Oak-dominated stands**

High fire frequency/intensity

**Pine-dominated stands**

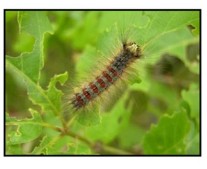

*Lymantria dispar* infestations

Oak mortality

Southern pine beetle infestations

Pine mortality

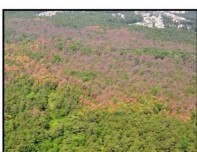

Pine seedling and sapling release
Pulse of pine regeneration

Oak release and regeneration (Uplands)
Hardwood release and regeneration (Lowlands)

Delaying succession

Accelerating succession

Moderate fire frequency/intensity

**Mixed composition, uneven age stands**

Moderate insect infestations

**Fig 1. A conceptual model of forest composition in intermediate age oak- and pine- dominated stands in the New Jersey Pinelands National Reserve.** In oak-dominated stands, repeated defoliation by *L. dispar* and differential mortality of oak trees and saplings facilitates the regeneration and release of pine seedlings and saplings. In pine-dominated stands, southern pine beetle causes significant pine tree mortality, while oaks in upland stands and other hardwoods in lowland stands are unaffected. These infestations are favoring the formation of uneven-aged, oak-pine mixedwoods in upland stands, and uneven-aged hardwood-pine mixedwoods in lowland stands.

measurements (e.g., [13, 30, 35]). In addition, several simulation models have captured the overall short-term dynamics of carbon and hydrologic cycling associated with insect infestations in these forests [27, 28, 36]. In summary, defoliators and bark beetles initially reduce the leaf area of susceptible species in infested stands by defoliation or host tree and sapling mortality, immediately reducing photosynthetic activity and autotrophic respiration, which decreases NEE and transpiration [13, 30, 37]. Compensatory photosynthesis by the remaining foliage, which is typically exposed to higher light levels, and the rapid cycling of nutrients from nutrient-rich litter and frass contribute to the maintenance of photosynthetic activity of the remaining foliage and facilitates resprouting of new foliage [38–41]. As a result, total $CO_2$ assimilation by photosynthesis, expressed as gross ecosystem productivity (GEP), evapotranspiration (Et) and ecosystem water use efficiency ($WUE_e$), defined as the amount of $CO_2$ assimilated per unit of water transpired, often recover relatively rapidly following insect damage or other disturbances [42–44].

Repeated defoliation over consecutive growing seasons, extensive bark beetle infestations, and other severe disturbances that result in tree and sapling mortality increase standing dead and coarse woody debris (CWD) on the forest floor. Additional detrital mass contributes to heterotrophic respiration as decomposition occurs, and has led to decadal-scale depressions in

annual NEP in some forests of the PNR [29, 30]. Following significant mortality of oaks resulting from repeated *L. dispar* defoliation of an oak-dominated stand, Renninger et al. [29] estimated that increased release of $CO_2$ from standing dead and CWD would depress NEP for up to two decades. Flux measurements at this site have documented that NEP has averaged only 0.4 T C ha$^{-1}$ yr$^{-1}$ over the decade following the peak of oak mortality, representing 22% of pre-infestation values in S1 Table [30]. Similarly, Xu et al. [36, 45] used repeated forest census plots documenting increases in CWD coupled with a process-based productivity model and reported that relatively low annual NEP occurred in oak-dominated stands that had been impacted by *L. dispar* in the Delaware Water Gap, Pennsylvania, USA.

In addition to increased standing dead and CWD in intermediate age forests, other long-term effects of insect infestations include changes in species composition and age class structure resulting from differential tree mortality and regeneration (Fig 1). How these longer-term changes in composition and structure potentially alter ecosystem functioning in forests of the mid-Atlantic region have not been explored in detail. To evaluate the conceptual model in Fig 1 and understand how compositional and structural changes could affect ecosystem functioning over decadal time scales, we characterized how the most recent infestations of *L. dispar* and southern pine beetle in the PNR have 1) altered forest composition and age class structure, and 2) how the resulting changes potentially affect NEP, evapotranspiration (Et), and WUEe. We used forest census data collected in plots based on FIA protocols pre- and post-infestation and in comparative insect-infested and control stands to characterize changes in forest composition and structure. Eddy covariance measurements of NEE, energy, and water vapor fluxes in intermediate age oak-dominated, oak-pine mixedwood, and pine-dominated stands were employed to quantify NEP, Et and WUEe pre-, during, and post-infestation.

## Materials and methods

### Site description

Research sites were located in Atlantic, Burlington, Cumberland, and Ocean Counties in the Pinelands National Reserve (PNR) of southern New Jersey, USA. Oak-dominated, mixed-composition, and pine-dominated stands comprise the upland forests, and lowland forests are dominated by pitch pine (*Pinus rigida* Mill.), mixed hardwoods, and Atlantic white cedar (*Chamaecyparis thyoides* (L.) B.S.P). Most stands have regenerated naturally following cessation of timber harvesting and charcoal production towards the end of the 19th century, and severe wildfires throughout the 20$^{th}$ century [12, 16, 18]. The climate is cool temperate, with mean monthly temperatures of 0.7 ± 2.4 and 24.6 ± 1.1°C in January and July, respectively (mean ± 1 SD;1988–2018; State Climatologist of New Jersey). Mean annual precipitation is 1183 ± 168 mm. Soils are derived from the Cohansey and Kirkwood formations, and upland soils are sandy, coarse-grained, and have low nutrient status, cation exchange capacity, and base saturation, while lowland soils are higher in accumulated organic matter and nutrients [46]. The landscape is characterized by a relatively high frequency of wildfires and prescribed burns compared to other forest ecosystems in the northeastern US; from 2004 to 2016, over 15,000 wildfires burned 36,654 ha and prescribed fires were conducted on 84,096 ha [18, 23, 47, 48]. On average, the annual area burned in prescribed fires now exceeds that burned in wildfires by a factor of two.

### *L. dispar* infestations and forest structure

*L. dispar* has defoliated primarily oaks in large areas of upland forest throughout southern New Jersey over the last two decades. From 2004 to 2016, total acreage with heavy (50 to 75%) and severe (> 75%) canopy defoliation has totaled 328,700 ha in the four counties studied

[49]. The majority of defoliation in a recent infestation occurred from 2005 to 2009, with peak damage occurring in 2007 when approximately 20% of upland forests in the PNR and 68,650 ha in the four studied counties were heavily to severely defoliated [13, 49].

Forest census plots based on FIA protocols [9] were sampled before (2004–2005), during (2007) and a decade after infestations (2018) to document the impacts of *L. dispar* infestations on forest composition and structure in three intermediate age stands of contrasting species composition in the PNR. Forest census plots were located in an oak-dominated stand at the Silas Little Experimental Forest (39.9156˚N, -74.5955˚E) in Brendan Byrne State Forest, a mixedwood stand co-dominated by pitch pine (*Pinus rigida* Mill.) and oaks at Fort Dix (39.9731˚N, -74.4341˚E), and a pitch pine-dominated stand near the Cedar Bridge fire tower (39.8398˚N, -74.3787˚E) in the Greenwood Wildlife Management Area, referred to below as "oak", "mixed" and "pine", respectively. Permission to access sites was granted through a long-term agreement between the USDA Forest Service and the New Jersey Department of Environmental Protection (NJDEP). Stands were selected to represent the dominant age class (75–95 years) of the three major upland forest types in the PNR, based on FIA data [50]. At the beginning of the study in 2004, the mean age of dominant trees was 90, 74 and 80 years at the oak, mixed and pine stands, respectively. The oak stand was dominated by chestnut oak (*Quercus prinus* L.), black oak (*Q. velutina* Lam.), white oak (*Q. alba* L.), and scarlet oak (*Q. coccinea* Muenchh.), with scattered shortleaf and pitch pines. The mixed stand was co-dominated by pitch pine and chestnut oak, with scattered white and post (*Q. stellata* Wangenh.) oaks. The pine stand was dominated by pitch pine, with post and white oak saplings in the lower canopy. All stands had bear and blackjack oaks (*Q. ilicifolia* Wang. and *Q. marilandica* Muench.), huckleberry (*Gaylussacia baccata* (Wang.) K. Koch and *G. frondosa* (L.) Torr. & A. Gray ex Torr.), and blueberry (*Vaccinium* spp.) in the understory. Sedges (*Carex pensylvanica* Lam.), bracken fern (*Pteridium aquilinum* (L.) Kuhn), mosses and lichens were also present. Further descriptions of each stand can be found in S2 Table and [13, 29, 30, 50].

Forest census measurements were conducted on five circular plots (201 m$^2$) located within 100 m of each eddy covariance tower (described below) that were sampled annually at the oak and pine stands, and periodically at the mixed stand (sampling details are in [13, 30, 51]). In addition, 1-km$^2$ grids consisting of 16 FIA-type plots in a 4 by 4 arrangement centered on each eddy covariance tower were sampled periodically [51], with plots that occurred in non-forested areas such as sand roads or fire breaks omitted from these analyses. During each census, species, diameter at breast height (DBH; 1.37 m), height, and crown condition were recorded for all live and dead trees (> 12.5 cm DBH) and saplings (2.5 to 12.5 cm DBH). Allometric equations were used to calculate aboveground biomass and biomass of foliage of trees and saplings (S2 Table; [52–54]). Censuses in the five 201 m$^2$ plots at each site were also used to monitor seedling and sapling recruitment and mortality. To estimate stem and foliage biomass of scrub oaks and shrubs in the understory, two to four 1.0 m$^2$ destructively harvested subplots adjacent to each 201 m$^2$ census plot were harvested during peak leaf area of each growing season, dried at 70˚C until dry and then weighed. Further descriptions of each stand can be found in S2 Table and [13, 29, 30].

## Southern pine beetle infestations and forest structure

The recent southern pine beetle outbreak in New Jersey started in approximately 2000, and by 2016, approximately 19,500 ha had been infested, resulting in mortality of pitch, shortleaf (*P. echinata* Mill.), and Virginia (*P. virginiana* Mill.) pines in pine-dominated stands [14, 24, 55]. Pitch pine dominated lowlands have been impacted to a greater extent than pine dominated upland stands [24].

Forest census plots based on FIA protocols [9] were installed in 10 uninfested and insect-damaged areas in untreated pine-dominated stands of intermediate age, as part of a 51-stand census of southern pine beetle damage conducted throughout the PNR in 2014 and 2015 [24]. Permission to access stands was granted by NJDEP and the appropriate state forest supervisors. Aerial and ground-based surveys conducted by New Jersey Department of Environmental Protection and Dartmouth College researchers were used to locate beetle-damaged areas on public lands (primarily state forests and wildlife management areas), which ranged in size from 0.5 to 35.0 ha and were sampled approximately two to five years following infestation by southern pine beetle [24]. Of the 51 stands, 10 stands were unmanaged and no southern pine beetle suppression activities were conducted in infested areas; census data from these stands were analyzed here because suppression treatments occasionally damaged remaining pine trees and saplings in infested areas [24]. In the remaining 41 stands, southern pine beetle suppression treatments consisted of felling infested trees and saplings and cutting a buffer around the infestation, and then either leaving pine stems in place ("cut and leave") or hauling logs to a landing zone and chipping them ("cut and chip"). All stands were initially dominated by pitch pine, with shortleaf and Virginia pine also present in some stands. The average age of sampled pine trees was 77 ± 24 years old (mean ± 1 SD) [25]. Upland stands also contained mixed oaks, sassafras (*Sassafrass albidum* (Nutt.) Nees), and an occasional beech (*Fagus grandifolia* Ehrh.) and lowland stands also contained red maple (*Acer rubrum* L.), black gum (*Nyssa sylvatica* Marshall), American holly (*Ilex opaca* Aiton), and sweetgum (*Liquidambar styraciflua* L). Further descriptions of each stand can be found in S3 Table and [24, 25].

Species, DBH, height, and crown position were recorded for all live and dead trees and saplings, and canopy cover was estimated visually for each FIA-type (168 $m^2$) subplot in infested and uninfested areas. Understory height, species composition, and visually estimated cover by species (including tree seedlings) were recorded for each subplot, and pine seedlings were tallied in each subplot when present. Allometric equations based on destructive harvests were used to estimate total aboveground biomass and biomass of foliage of pine trees and saplings in each hsubplot (S3 Table; [24, 53]). Published values were used to estimate biomass and biomass of foliage for oaks and other hardwoods [52, 54, 56].

## Leaf area and foliar nitrogen content

Specific leaf area (SLA; $m^2$ g dry weight$^{-1}$) of foliage of the dominant canopy and understory species was measured with a leaf area meter (LI-3000a, LI-COR Inc., Lincoln, Nebraska, USA) and a conveyer belt (LI-3050c, LI-COR Inc.) using fresh samples of leaves or needle fascicles, which were then dried at 70˚C and weighed. Canopy leaf area index (LAI; $m^2$ $m^{-2}$ ground area) was estimated by multiplying leaf or needle mass calculated from allometric equations for each species by the appropriate SLA value and then summing results for all species. Projected leaf area of pine needle fascicles was multiplied by π/2 to calculate one-sided LAI. Understory LAI at the oak, mixed and pine stands was estimated by multiplying foliage mass of shrubs and oaks obtained from harvested 1.0 $m^2$ plots by the corresponding SLA values. Litterfall was collected monthly at the oak, mixed and pine stands when present from two 0.4-$m^2$ wire baskets per plot and used to estimate foliage mass and area for periods when extensive defoliation occurred. Relationships between leaf litter mass and SLA were developed for the dominant species using the same protocol used for fresh foliage.

Canopy and understory foliage was sampled for nitrogen concentrations ([N]) at the time of peak leaf area during the growing season at the oak, mixed and pine stands throughout the study. Oven-dry samples of live leaves or needles were ground using a Wiley mill (Thomas Scientific, Swedesboro, NJ, USA) and digested along with appropriate standards

using a modified Kjeldahl method [57]. An Astoria 2 Analyzer (Astoria-Pacific International, Clackamas, OR, USA) was used to measure the ammonium concentration of each sample, and results were converted to [N] in foliage samples. Values for [N] of foliage were consistent with those reported in Renninger et al. [58, 59] and Guerrieri et al. [43, 44] for foliage at the oak and pine stands. Nitrogen content (g N m$^{-2}$ ground area) in canopy and understory foliage of each dominant species was then calculated by multiplying species-specific [N] by corresponding estimates of foliar biomass. At the stands infested by southern pine beetle, N content of foliage also was estimated using needle or leaf biomass estimates and mean foliar [N] of the dominant species.

## Ecosystem functioning of oak-, mixed and pine-dominated stands

Closed-path eddy covariance systems and meteorological sensors mounted on antenna towers were used to quantify net ecosystem exchange of $CO_2$ (NEE) and latent heat flux at the oak-, mixed and pine-dominated stands. Values were integrated over the appropriate time intervals to estimate daily and annual net ecosystem production (NEP), evapotranspiration (Et), and ecosystem water use efficiency (WUE$_e$) pre-, during and post-infestation by *L. dispar*. Near-continuous measurements commenced in 2004 at the oak stand (two years prior to *L. dispar* infestations) and in 2005 at the mixed and pine stands (one and two years prior to *L. dispar* infestations, respectively). Eddy covariance systems, meteorological sensors, and data processing methods are described in detail in Clark et al. [13, 30] and in S4 Table. In summary, half-hourly fluxes were calculated from raw 10 Hz flux data using EdiRE [60], and values were rejected when instrument malfunction occurred, during measurable precipitation or when icing occurred, and when friction velocity (u*) < 0.2 m s$^{-1}$, which ensured well-mixed conditions. To estimate half-hourly NEE values when we did not have measurements, daytime NEE was modeled by fitting a rectangular hyperbola to the relationship between photosynthetically active radiation (PAR) and NEE at bi-weekly (May) to 3-month (summer; June 1 –August 31) periods. Nighttime NEE was modeled by regressing half-hourly net exchange rates on air temperature using an exponential function. Model parameters and their error terms for the relationships between half-hourly daytime or nighttime NEE and meteorological variables were calculated using SigmaPlot software (Version 12.5, Systat Software, Inc., San Jose, CA, USA). Continuous meteorological data and the appropriate model were then used to fill gaps for periods when fluxes were not measured, and measured and modeled values were summed to estimate daily and annual NEP. Ecosystem respiration (R$_e$) was estimated using nighttime NEE and continuous half-hourly air temperature during the growing season and soil temperature during the dormant season. Error in gap-filling NEE and R$_e$ was evaluated for daytime and nighttime data using ± 1 standard error (SE) of each parameter used to model half-hourly NEE (see [30] for details). Daily NEP and R$_e$ were summed to estimate daily and annual GEP values for each stand.

Evapotranspiration was estimated from latent heat fluxes calculated using EdiRE. Meteorological measurements were used to calculate available energy, defined as net radiation–(soil heat flux + heat storage terms), and gaps in Et data were filled using linear functions in SigmaPlot [37]. Half-hourly Et estimates were then summed to calculate daily and annual values. Ecosystem water use efficiency (g C kg H$_2$O$^{-1}$) is defined here as the ratio of daily GEP to transpiration [42–44]. Following Clark et al. [42], we used data collected when we assumed the canopy was dry to maximize the contribution of transpiration to Et in these calculations, and days with recorded precipitation and the day following each rain event when precipitation $\geq$ 10 mm day$^{-1}$ were excluded from analyses.

## Statistical analyses

All datasets were first tested for normality using Kolmogorov-Smirnov tests, and homogeneity of variances among groups were tested using Levene's test. Values for BA and aboveground biomass of trees and saplings, leaf area index, and nitrogen content of foliage among stands infested by *L. dispar* were compared using ANOVA analyses. Comparisons among stands were made with Tukey´s Honestly Significant Difference (HSD) tests that adjusted significance levels for multiple comparisons. Paired sample T-tests were used to compare pre- and post-infestation values within stands. Paired-sample T-tests were also used to compare forest structure variables in infested and uninfested areas in stands that had been impacted by southern pine beetle. Half-hourly values of NEE for daytime and nighttime periods, and daily values of NEP, Et and WUE$_e$ were compared among stands using ANOVA analyses. Because half-hourly and daily values were not independent, we randomly selected 25 subsets consisting of 25 values for each variable, and then tested for differences among stands or time periods [42]. Comparisons among stands were made with Tukey´s HSD tests. All statistical analyses were conducted using SYSTAT 12 software (Systat Software, Inc., San Jose, CA, USA).

# Results

## *L. dispar* infestations and forest structure

Prior to *L. dispar* infestations, BA of trees and saplings was similar at the oak, mixed, and pine stands (Fig 2A and Table 1), while aboveground tree biomass was greater at the oak stand than at the mixed and pine stands (S2 Table). Leaf area was greatest at the oak stand and least at the pine stand during the growing season (Fig 2B). Similarly, N content of foliage was greater at the oak stand than at the pine stand during the growing season, with foliage of oak trees and saplings containing 77%, 51% and 12% of total foliar N content at the oak, mixed and pine stands, respectively (Fig 2C and Table 1). Standing dead tree and saplings and coarse woody debris mass was < 3.1 ± 0.6 t ha$^{-1}$ at the three stands at the beginning of the study (S2 Table; see [30] for details).

Infestations of *L. dispar* occurred at the oak stand during the growing seasons of 2006 to 2008. During the peak of defoliation in 2007, leaf area of oaks, pines and understory vegetation was reduced to near zero, and a second partial leaf-out resulted in a total leaf area and foliar N content of only 42% and 40% of pre-defoliation values, respectively ("Inf" in Fig 2B and 2C). Following infestations, oak mortality peaked from 2009 to 2011, and by 2018 oak tree and sapling BA had been reduced by ≈ 40% compared to pre-infestation values. Overstory mortality resulted in the release of pine saplings and establishment of seedlings in the understory, and by 2018 pine trees and saplings accounted for 38% of total BA ("Post" in Fig 2A). Although BA increment of surviving trees at the oak stand was positive, total BA and above-ground biomass were similar at the beginning and end of the study in 2018. Oak mortality reduced stand leaf area and foliar N content, and in 2018, N content of oak tree and sapling foliage was less than pre-infestation values (T$_4$ = 3.53, P < 0.05), accounting for only 63% of total foliar N content (Fig 2C and Table 2). Oak mortality following *L. dispar* infestations resulted in a maximum standing dead and CWD mass of 31.1 ± 9.1 t ha$^{-1}$ (mean ± 1 SE) in 2011, and by 2018 standing dead and CWD mass was estimated to be 19.0 ± 5.3 t ha$^{-1}$ (S2 Table; see [30] for details).

At the mixed stand, *L. dispar* infestations occurred from 2006 to 2008, but in contrast to the oak stand, oak tree and sapling mortality was minor following infestations (Fig 2A). Defoliation by *L. dispar* reduced leaf area and foliar N content of deciduous species to very low values in 2007 but had relatively little effect on foliage of pine trees and saplings (Fig

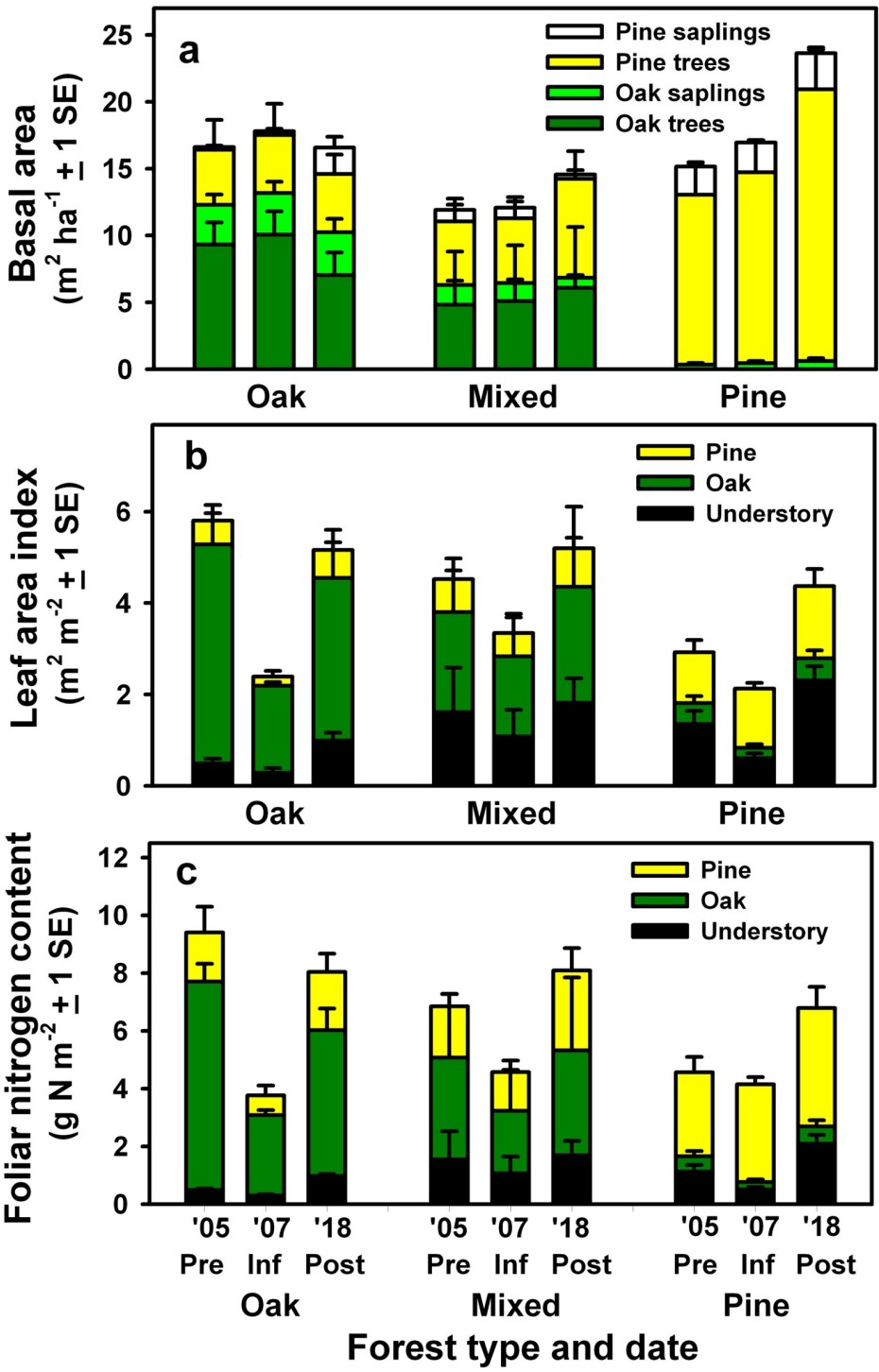

**Fig 2. Effects of *L. dispar* infestations on forest composition and structure.** (A) Basal area of pine and oak trees and saplings, (B) maximum leaf area of pines, oaks and understory vegetation during the growing season, and (C) maximum nitrogen content in foliage of pines, oaks and understory vegetation during the growing season prior to infestations in 2004 (Pre), during the year of peak defoliation in 2007 (Inf), and a decade following infestations in 2018 (Post) at the oak-, mixed and pine-dominated stands. Pine tree, sapling and seedling leaf area is expressed as one-sided LAI. Leaf area and foliar nitrogen content during infestations in 2007 reflect values after a second leaf-out of foliage in mid-July 2007 following complete defoliation of the oak stand, and partial defoliation of the mixed and pine stands.

**Table 1. Results of ANOVA and Tukey's HSD analyses for structural characteristics of forests infested by *L. dispar*.**

| Comparison | $F_{2,12}$ | P | Contrasts | Figure |
|---|---|---|---|---|
| **Before *L. dispar* infestations in 2004 ("Pre" in Fig 2)** | | | | |
| Tree and sapling BA | 1.0 | NS | NS | 2A |
| Leaf area index | 4.6 | < 0.05 | O > P | 2B |
| N content of foliage | 4.4 | < 0.05 | O > P | 2C |
| **During *L. dispar* infestation in 2007 ("Inf" in Fig 2)** | | | | |
| Tree and sapling BA | 2.3 | NS | NS | 2A |
| Leaf area index | 1.6 | NS | NS | 2B |
| N content of foliage | 0.2 | NS | NS | 2C |
| **Following *L. dispar* infestations in 2018 ("Post" in Fig 2)** | | | | |
| Tree and sapling BA | 3.6 | < 0.10 | P > M | 2A |
| Leaf area index | 0.2 | NS | NS | 2B |
| N content of foliage | 0.4 | NS | NS | 2C |

Statistical tests are for tree and sapling basal area (BA), and canopy and understory leaf area index and nitrogen (N) content of foliage in stands before, during and following *L. dispar* infestations shown in Fig 2. O = oak stand, M = mixed stand, P = pine stand. NS = not significant.

2B). By the end of the study in 2018, BA of trees and saplings had increased by 22% compared to values in 2004. Increases in both pine and oak tree BA resulted from growth increments and sapling recruitment, despite some sapling mortality that occurred during the three prescribed fires conducted between 2006 and 2018. Leaf area and N content of foliage during the growing season of 2018 at the mixed stand had increased 15% and 18% compared to 2004, although increases were not statistically significant (Fig 2B and 2C and Table 1).

At the pine stand, oak sapling mortality was minimal following *L. dispar* infestations (Fig 2A). Partial defoliation of the understory and oak saplings by *L. dispar* in 2007 reduced understory LAI and N content compared to pre-disturbance periods but had little effect on pine foliage (Fig 2B and 2C). Growth increments of trees and saplings resulted in an increase in BA of 56% between 2004 and 2018. Although prescribed fires were conducted at the pine stand in 2008 and 2013 (see [30] for details), leaf area and foliar N content had increased 50% and 49% by 2018 when compared to 2004, following a longer-term trend of recovery from a severe wildfire that had occurred in 1995 (Fig 2B and 2C).

**Table 2. Results of paired-sample T-tests for structural characteristics of stands infested by southern pine beetle.**

| Comparison | $T_{1,9}$ | P value | Figure |
|---|---|---|---|
| **Tree and sapling BA** | 6.0 | < 0.01 | 3A |
| Pine trees | 6.9 | < 0.01 | 3A |
| Pine saplings | 0.2 | NS | 3A |
| Oaks and other hardwoods | 1.1 | NS | 3A |
| **Leaf area index** | 5.2 | < 0.01 | 3B |
| Pine trees and saplings | 7.7 | < 0.01 | 3B |
| Oaks and hardwoods | 0.4 | NS | 3B |
| **N content of foliage** | 6.6 | < 0.01 | 3C |
| Pine trees and saplings | 7.7 | < 0.01 | 3C |
| Oaks and other hardwoods | 0.6 | NS | 3C |

Statistical tests are for trees and saplings in infested and uninfested areas shown in Fig 3.

## Southern pine beetle infestations and forest structure

Pine tree basal area averaged 21.8 ± 2.8 m$^2$ ha$^{-1}$ in areas that were not infested by southern pine beetle in southern New Jersey. Pine trees and saplings in uninfested areas accounted for 75% of total BA, 61% of tree and sapling leaf area and 78% of tree and sapling foliar N content (Fig 3 and S3 Table). Infestations of southern pine beetle resulted in significant mortality of pitch, shortleaf and Virginia pine trees, averaging 92% of pine tree BA, while pine sapling BA was reduced by only approximately 5% in infested areas (Fig 3A). Beetle infestations had little effect on the BA of oak trees and saplings in upland areas or of other hardwood trees and saplings such as red maple and black gum in lowland areas (Fig 3A and Table 2).

Following southern pine beetle infestations, tree and sapling leaf area and foliar N content in infested areas averaged 42% and 26% of values for uninfested areas, respectively (Fig 3B and 3C). While pine leaf area and foliar N content was reduced significantly, leaf area and foliar N content of oaks and other hardwoods were nearly unchanged (Fig 3B and 3C and Table 2). CWD was highly variable in infested areas due to a large proportion of standing dead trees in some stands (S3 Table).

## Convergence of forest structure following insect infestations

By the end of the study, changes in stand composition and structure at the oak-dominated stand impacted by *L. dispar* and at untreated, previously pine-dominated stands infested by southern pine beetle converged on attributes characterizing the mixed stand measured at the beginning of the study. For example, Fig 4 indicates the similarity in relative BA of trees and saplings among the oak stand in 2018 following *L. dispar* defoliation, the mixed stand at the beginning of the study in 2005, and untreated pine stands following infestation by southern pine beetle.

## Ecosystem functioning of oak-, mixed and pine-dominated stands

Summertime (June 1 to August 31) half-hourly NEE during midday clear sky conditions and daily (24-hour) NEP were greater at the oak stand than at the mixed and pine stands before *L. dispar* infestations (Tables 3 and 4 and Fig 5A). However, the opposite pattern occurred during the spring and fall seasons; before leaf expansion of oaks and understory vegetation in spring (April to mid-May), half-hourly NEE during midday clear sky conditions and daily NEP were greater at the pine stand than at the mixed and oak stands (Fig 5A and Table 4). Annual NEP was similar at the oak and pine stands, and somewhat lower at the mixed stand before *L. dispar* infestations, although complete annual data for the pre-disturbance period at the mixed stand were only available for 2005 (Table 5). Daily GEP and WUE$_e$ were also greater at the oak stand than at the mixed and pine stands during the summer, while daily GEP and WUE$_e$ during the spring were greater at the pine stand than at the oak and mixed stands (Figs 5B and 6B). Daily evapotranspiration rates during the summer were similar among stands, with annual values averaging 51% to 62% of incident precipitation (Table 5).

Changes in the distribution of leaf area and foliar N content at the oak stand following *L. dispar* infestations coincided with springtime increases in half-hourly NEE during midday when PAR > 1500 μmol m$^{-2}$ s$^{-1}$ and daily NEP, and reduced summertime half-hourly midday NEE and daily NEP, with values during both periods approaching those previously measured at the mixed stand at the beginning of the study (Table 3 and Fig 5A; post-defoliation values in 2018). Daily GEP during the summer at the oak stand was similar in 2005 and 2018, but daily WUE$_e$ was somewhat lower in 2018 and equivalent to rates measured previously at the mixed stand in 2005 (Figs 5 and 6). In contrast, seasonal patterns of daily NEP, GEP, Et and WUE$_e$ at the pine stand were similar in 2005 and 2018 (Figs 5 and 6).

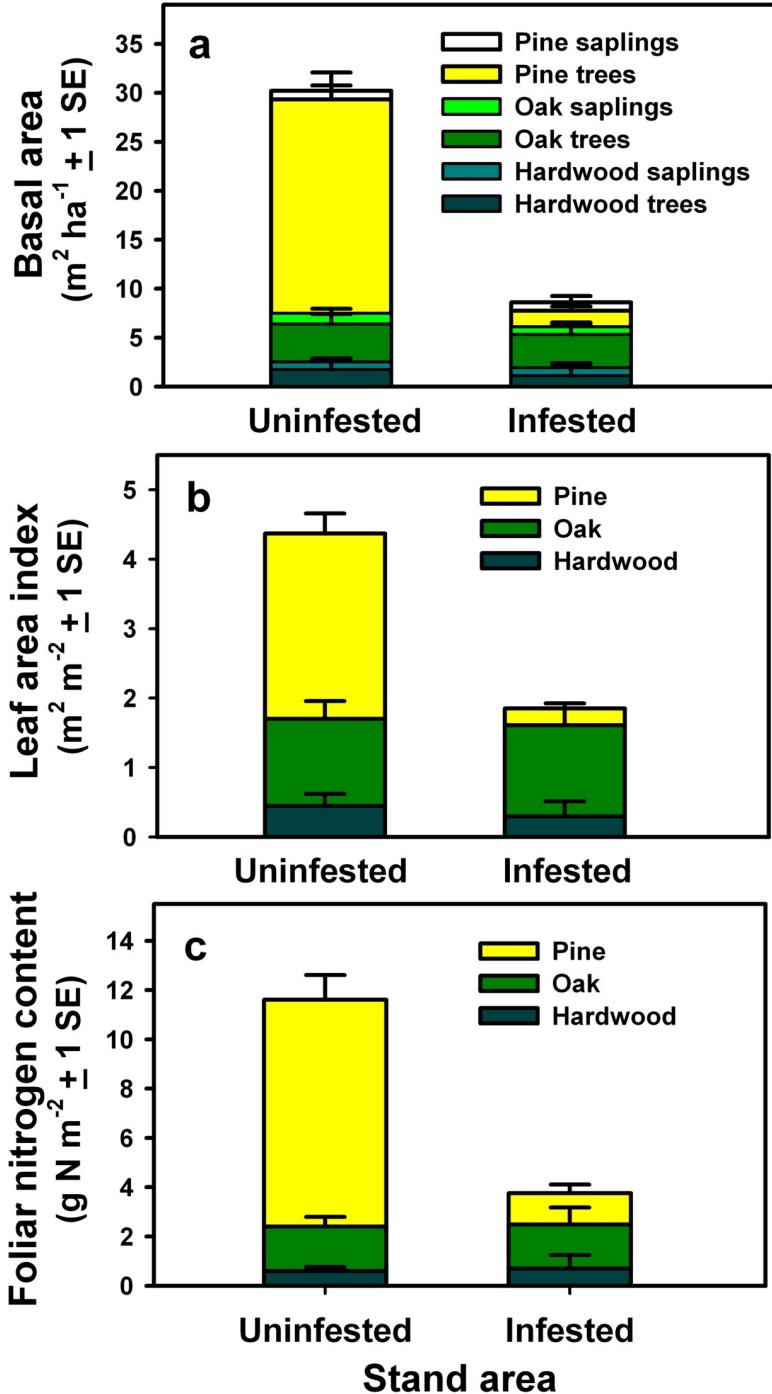

**Fig 3. Effects of southern pine beetle on forest composition and structure.** (A) Basal area of pine, oak and other hardwood trees and saplings, (B) maximum leaf area of pines, oaks, and other hardwoods during the growing season, and (C) maximum foliar N content of pines, oaks, and other hardwoods during the growing season in uninfested areas and areas following infestation of southern pine beetle in southern New Jersey. Other hardwoods include red maple (*Acer rubrum* L.), black gum (*Nyssa sylvatica* Marshall), sassafras (*Sassafras albidum* (Nutt.) Nees), sweet gum (*Liquidambar styraciflua* L.), and sweetbay magnolia (*Magnolia virginiana* L.).

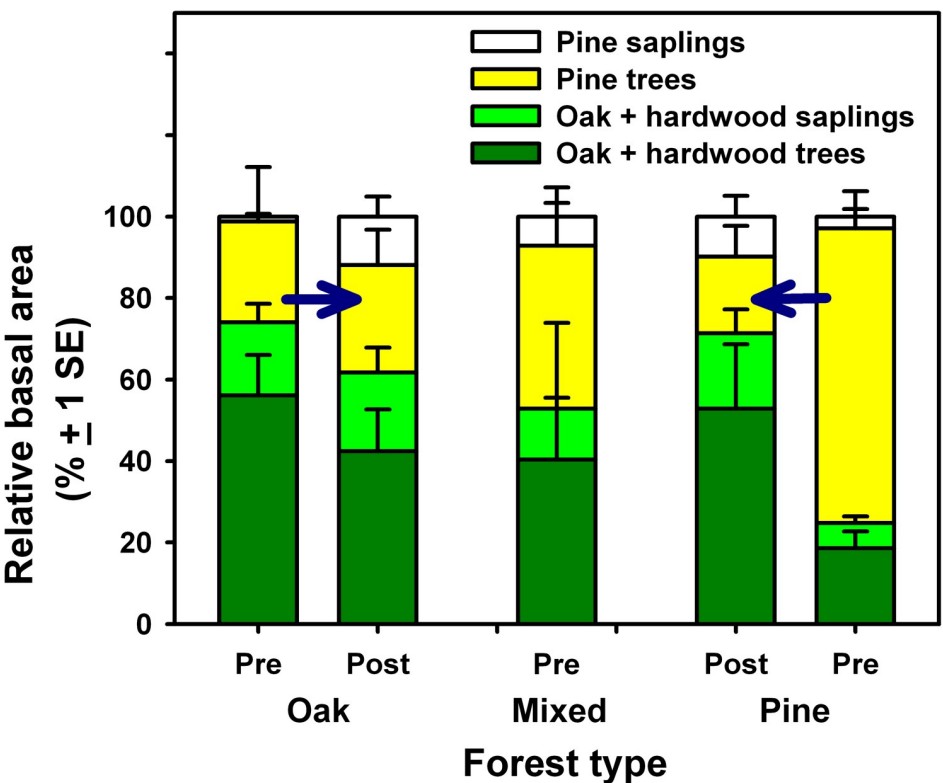

**Fig 4. Relative basal area of pine and hardwood trees and saplings.** Data are from the oak stand before *L. dispar* infestation in 2005 and following tree and sapling mortality in 2018, the mixed stand at the beginning of the study in 2005, and untreated pine-dominated areas following infestation by southern pine beetle and uninfested areas. Oaks and other hardwoods have been combined as "hardwoods". Arrows indicate the directional changes caused by insect infestations.

## Discussion

Infestations of *L. dispar* are delaying successional changes in oak-dominated stands and southern pine beetle infestations are accelerating changes in pine-dominated stands, while having only moderate effects in mixedwood stands on the mid-Atlantic Coastal Plain. In our study, the composition and structure of oak-dominated stands infested by *L. dispar* and of pine-dominated stands infested by southern pine beetle are converging on those characterizing oak-pine mixedwoods in upland stands and hardwood-pine mixedwoods in lowland stands, with similarly proportioned distributions of BA, leaf area, and foliar N content among oaks or other hardwoods and pines. In the long term, repeated but less severe insect infestations and current fire management strategies, including both wildfire suppression and the extensive use of prescribed fires, will likely favor the persistence of oak-pine and hardwood-pine mixedwoods throughout the PNR, consistent with the conceptual model in Fig 1. These outcomes parallel observations in other mixedwood forests consisting of species with varying susceptibility to insects, which can persist through time because of greater associational resistance to infestations compared to those dominated by a single species or genus [61, 62]. They are also consistent with the theoretical framework proposed by Kern et al. [22], who predicted that insect infestations, a disturbance from above because the canopy is impacted, coupled with low-intensity surface fires, a disturbance from below which promotes the regeneration of shade-intolerant species, would result in the persistence of mixedwood forests through time. Our

**Table 3. Half-hourly net $CO_2$ exchange (NEE) during spring (April 1 to May 15) and summer (June 1 to August 31) months at the oak, mixed and pine stands.**

| Season | Half-hourly NEE ($\mu$mol $CO_2$ m$^{-2}$ s$^{-1}$) | | | |
|---|---|---|---|---|
| | Oak | Mixed | Pine | Statistics |
| **Daytime, before *L. dispar* infestations** | | | | |
| Spring | 0.42 ± 1.68[a] | -2.57 ± 2.01[b] | -7.05 ± 1.69[c] | $F_{2,72}$ = 109.0, P < 0.001 |
| Summer | -19.80 ± 4.65[a] | -16.18 ± 3.84[b] | -15.92 ± 3.73[b] | $F_{2,72}$ = 7.0, P < 0.01 |
| **Nighttime, before *L. dispar* infestations** | | | | |
| Spring | 2.32 ± 1.63 | 1.87 ± 1.52 | 2.38 ± 1.64 | $F_{2,72}$ = 0.8, NS |
| Summer | 5.86 ± 2.65[ab] | 4.80 ± 1.91[a] | 6.51 ± 2.25[b] | $F_{2,72}$ = 3.6, P < 0.05 |
| **Daytime, during *L. dispar* infestation in 2007** | | | | |
| Spring | 0.74 ± 1.91[a] | -2.88 ± 1.78[b] | -6.28 ± 2.24[c] | $F_{2,72}$ = 103.9, P < 0.001 |
| Summer | -2.40 ± 2.96[a] | -6.72 ± 5.69[b] | -10.08 ± 3.34[c] | $F_{2,72}$ = 29.4, P < 0.001 |
| **Nighttime, during *L. dispar* infestation in 2007** | | | | |
| Spring | 1.61 ± 1.46 | 1.72 ± 1.00 | 2.32 ± 1.63 | $F_{2,72}$ = 2.4, NS |
| Summer | 3.54 ± 2.54[a] | 4.01 ± 1.91[ab] | 5.19 ± 2.40[b] | $F_{2,72}$ = 4.4, P < 0.05 |
| **Daytime, following *L. dispar* infestations in 2018** | | | | |
| Spring | -1.24 ± 2.14[a] | --- | -8.11 ± 3.23[b] | $T_{48}$ = 8.9, P < 0.01 |
| Summer | -15.90 ± 5.03 | --- | -14.13 ± 3.55 | $T_{48}$ = 1.4, NS |
| **Nighttime, following *L. dispar* infestations in 2018** | | | | |
| Spring | 2.88 ± 2.21 | --- | 2.57 ± 2.20 | $T_{48}$ = 0.5, NS |
| Summer | 6.44 ± 3.57 | --- | 6.19 ± 3.25 | $T_{48}$ = 0.3, NS |

Daytime NEE values are midday values at PAR $\geq$ 1500 $\mu$mol m$^{-2}$ s$^{-1}$ and nighttime NEE values are during well-mixed conditions when friction velocity (u*) is $\geq$ 0.2 m s$^{-1}$. All values are means ± 1 SD. NS = not significant.

**Table 4. Results of ANOVA analyses for ecosystem functioning of the oak, mixed and pine stands.**

| Comparison | $F_{2,72}$ | P | Contrasts | Figure |
|---|---|---|---|---|
| **Before *L. dispar* infestations in 2005** | | | | |
| Spring NEP | 68.4 | < 0.001 | O = M < P | 5A |
| Summer NEP | 8.6 | < 0.001 | O > M = P | 5A |
| Spring GEP | 42.3 | < 0.001 | O = M < P | 5B |
| Summer GEP | 10.6 | < 0.001 | O > M = P | 5B |
| Spring Et | 25.0 | < 0.001 | O = M < P | 6A |
| Summer Et | 1.0 | NS | NS | 6A |
| Spring WUE$_e$ | 21.4 | < 0.001 | O = M < P | 6B |
| Summer WUE$_e$ | 14.0 | < 0.001 | O > M = P | 6B |
| **Following *L. dispar* infestations in 2018** | | | | |
| Spring NEP | 29.7 | < 0.001 | O = M < P | 5A |
| Summer NEP | 2.8 | NS | NS | 5A |
| Spring GEP | 19.4 | < 0.001 | O = M < P | 5B |
| Summer GEP | 7.0 | < 0.005 | O > M = P | 5B |
| Spring Et | 14.2 | < 0.001 | O = M < P | 6A |
| Summer Et | 2.2 | NS | NS | 6A |
| Spring WUE$_e$ | 22.1 | < 0.001 | O = M < P | 6B |
| Summer WUE$_e$ | 1.9 | NS | NS | 6B |

Statistical tests are for daily net ecosystem production (NEP), gross ecosystem production (GEP), evapotranspiration (Et), and ecosystem water use efficiency (WUE$_e$) at the oak, mixed and pine stands shown in Fig 5. O = oak stand, M = mixed stand, P = pine stand, NS = not significant.

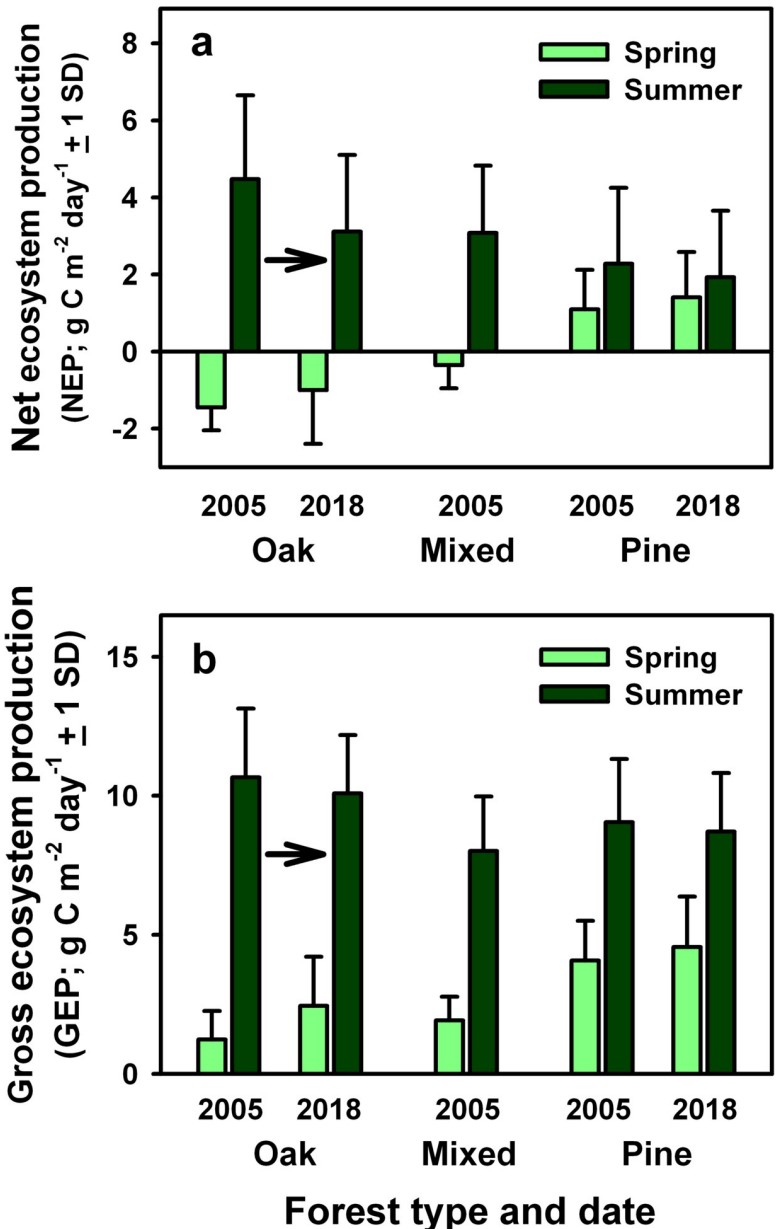

**Fig 5. Productivity of the oak, mixed, and pine stands before *L. dispar* infestations in 2005 and following *L. dispar* infestations in 2018.** Data are presented for (A) daily net ecosystem production, (B) daily gross ecosystem production during late spring (April 1 to May 15) and summer (June 1 to August 31) months. Arrows indicate the directional changes in forest structure and composition following *L. dispar* infestations. Pre-infestation data are adapted from Clark et al. [13, 42].

study further suggests that ecosystem functioning, especially NEP and GEP, will recover relatively rapidly in oak–pine or other hardwood–pine mixedwood stands, as they can be expected to experience less defoliation and/or lower amounts of tree and sapling mortality compared to infestations of *L. dispar* in oak-dominated stands or southern pine beetle in pine-dominated stands.

Tree species composition and initial foliage quality of canopy species (approximated by foliar [N] in our study) influence the occurrence of the multi-year population outbreaks of *L.*

**Table 5. Annual values of net ecosystem production (NEP), gross ecosystem production (GEP), precipitation, and evapotranspiration (Et) at the oak, mixed and pine stands.**

| Stand | NEP | GEP | Precipitation | Et |
|---|---|---|---|---|
| | g C m$^{-2}$ yr$^{-1}$ | g C m$^{-2}$ yr$^{-1}$ | mm yr$^{-1}$ | mm yr$^{-1}$ |
| **Before *L. dispar* infestations in 2005** | | | | |
| Oak | 169 ± 24 | 1593 ± 58 | 1100 | 647 |
| Mixed | 137 ± 19 | 1205 ± 57 | 1184 | 607 |
| Pine | 173 ± 18 | 1513 ± 36 | 1230 | 757 |
| **During *L. dispar* infestation in 2007** | | | | |
| Oak | -246 ± 14 | 676 ± 46 | 934 | 442 |
| Mixed | -20 ± 20 | 958 ± 52 | 1135 | 419 |
| Pine | 49 ± 7 | 1378 ± 43 | 1052 | 593 |
| **Following *L. dispar* infestations in 2018** | | | | |
| Oak | 27 ± 15 | 1550 ± 43 | 1397 | 740 |
| Pine | 173 ± 18 | 1585 ± 48 | 1580 | 858 |

Data are for years before, during, and following *L. dispar* infestations. Error terms were calculated from maximum deviations from average values generated using ± 1 SE of parameter values used to gap-fill missing half-hourly daytime and nighttime NEE (see [13, 30] for complete description of gap-filling procedures and error term calculations).

*dispar* which result in the extensive mortality of susceptible species [7, 63–65]. In our study, differential mortality of black and white oaks, which have relatively high foliar [N] (≈ 2.1% N; [66, 67]), resulted in increased dominance of chestnut oak (as reflected in increased relative BA) with lower foliar [N] (≈ 1.9% N; [43, 44]). Reduced cover of oak trees and saplings facilitated the growth of pine saplings and the establishment and recruitment of pine seedlings, which have much lower foliar [N] than canopy oaks (1.0 to 1.3% N), and increased leaf area and biomass of understory vegetation [30]. The decrease in BA of susceptible oak species and reduction in oak leaf area, combined with lower mean [N] of canopy foliage because of the increase in pine foliage, will likely reduce the severity of *L. dispar* infestations in the future [7, 11, 63–65]. This outcome is consistent with observations from oak-pine mixedwood stands, where although infestations occurred and oak trees and saplings were defoliated, cumulative mortality was less extensive. Over time, repeated but less severe insect damage to oaks coupled with pulses of pine seedling establishment and saplings recruitment associated with prescribed fires will delay successional changes and likely result in uneven age mixedwood stands, as proposed in Fig 1.

Extensive pine tree mortality in areas infested by southern pine beetle reported here is similar to their impacts in pine-dominated forests across the southeastern USA [68]. Initial BA of pine trees and saplings in infested stands in the PNR (≈ 22.7 m$^2$ ha$^{-1}$) was greater than the average BA that can favor the large southern pine beetle aggregations leading to significant pine tree mortality in southeastern USA forests (≈ 18 m$^2$ ha$^{-1}$) [14, 25, 68]. In contrast, oak trees and saplings in upland stands and other hardwood trees and saplings in lowland stands were essentially unaffected in infested areas, and they were often retained where suppression treatments (e.g., cut and leave, cut and chip) were conducted in the PNR [24], and more recently, further north on the Atlantic Coastal Plain on Long Island, New York, USA [15, 55]. Southern pine beetle rarely impacted pines in oak-dominated stands in upland locations, or in hardwood-dominated stands in lowland locations in the PNR. Similarly, Huess et al. [15] reported that pine mortality was lower in mixed pine–oak stands than in pine-dominated stands following southern pine beetle infestations on Long Island, NY. In our study, BA of pine trees and saplings (≈ 2.5 m$^2$ ha$^{-1}$) in infested and treated areas was well below the

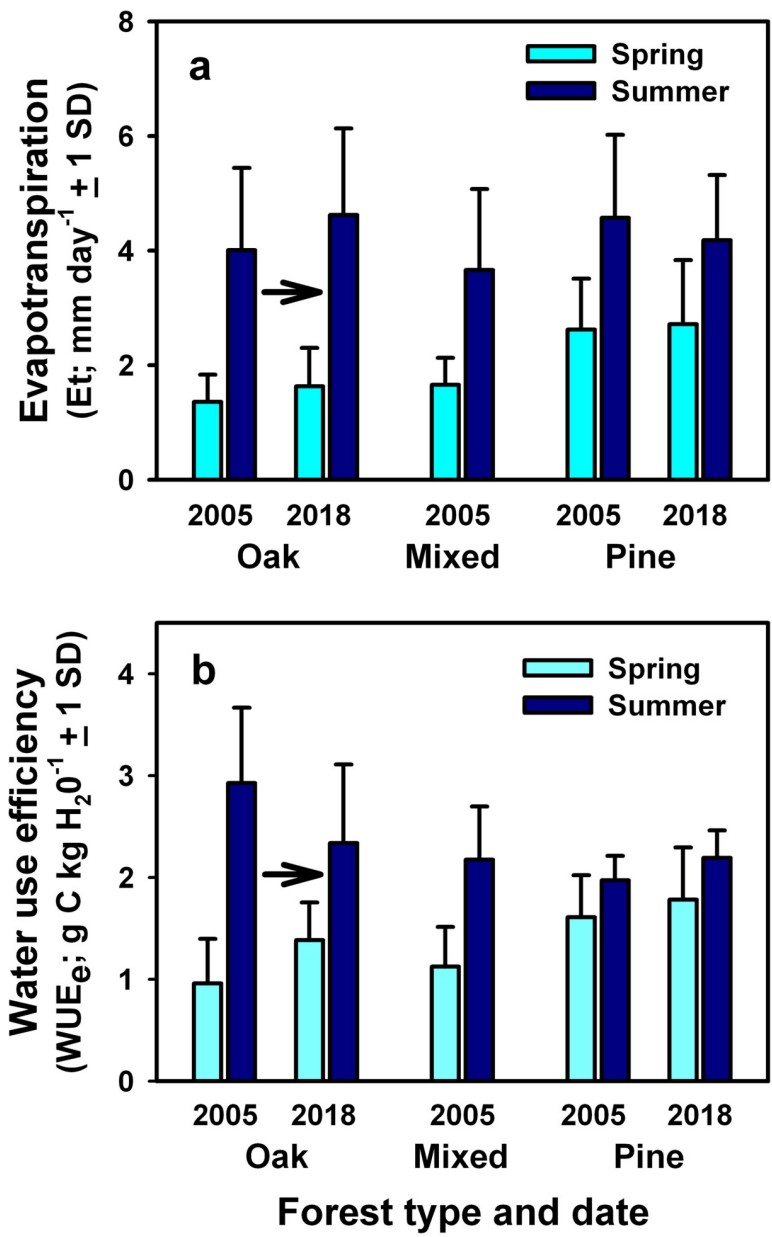

**Fig 6. Evapotranspiration and water use efficiency of oak, mixed, and pine stands before *L. dispar* infestations in 2005 and following *L. dispar* infestations in 2018.** Data are presented for (A) daily evapotranspiration, and (B) daily ecosystem water use efficiency during late spring (April 1 to May 15) and summer (June 1 to August 31) months. Arrows indicate the directional changes in forest structure and composition following *L. dispar* infestations. Pre-infestation data are adapted from Clark et al. [37, 42].

densities that would support future aggregations of southern pine beetles [24, 55, 68]. Overall, southern pine beetle damage can accelerate succession in infested stands on the Atlantic Coastal Plain, also resulting in the formation of uneven age, mixedwood stands, consistent with the conceptual model in Fig 1.

Field measurements and model simulations indicate that daily NEP, GEP and WUEe during the growing season are greatest in oak-dominated stands and daily values in oak-pine mixedwood stands are intermediate between oak- and pine-dominated stands [13, 30]. Daily NEP

and GEP during the growing season are strongly correlated with leaf area and canopy N content in forests of the PNR [30, 31, 59, 67], consistent with the relationship between LAI, canopy N content, and NEP during the growing season reported for forests at landscape to regional scales throughout the Northeastern USA [e.g., 35, 69, 70]. Pines and other evergreens in mixedwood and pine-dominated stands, however, are more productive during periods of time when deciduous oaks, other deciduous hardwoods, and many understory species are dormant. Integration of the seasonal patterns of C assimilation by oaks, pines and understory species results in more similar annual rates of NEP, GEP and WUEe among oak-dominated, oak-pine mixedwood, and pine-dominated stands [13, 30; S1 Table]. Thus, the long-term changes in species composition and structure associated with insect infestations may have little effect on forest carbon dynamics and hydrologic cycling at annual time scales in forests of the Mid-Atlantic region. In contrast, short-term carbon dynamics following infestations are strongly influenced by stand species composition. Field measurements and model simulations have documented how insect-driven disturbance and widespread tree and sapling mortality of susceptible species can reduce NEP for at least a decade following infestations [26–30, 36, 45]. Large-scale assessments have documented how differential mortality of oaks caused by *L. dispar* infestations in oak-hickory forests have reduced or negated net increases in BA and aboveground biomass across the mid-Atlantic region [7, 65, 71]. Because mixedwood stands are more resistant to infestations and sustain less extensive damage, they will likely maintain continuity in ecosystem functioning to a greater extent than oak- or pine-dominated stands during and following insect infestations [61, 62].

Numerous forest tree species in the mid-Atlantic region are tolerant of drought and fire, and many are characterized by regeneration strategies that enhance survival following fires or other disturbances (e.g., epicormic budding in pitch and shortleaf pines, serotinous cones in some pitch pine populations, prolific resprouting in most oaks and red maple) [16, 38, 72]. The use of prescribed fire to promote the regeneration of both oaks and pines is well documented in oak–pine mixedwood stands throughout the mid-Atlantic region [47, 73–77]. In the PNR, the majority of prescribed fires are conducted during the early spring, before oaks and other hardwoods have leafed out, and when pitch and shortleaf pines carry only a single cohort of needles. A seasonal peak in severe wildfires follows later in spring, also occurring before leaf expansion of deciduous species [23, 47, 73, 78]. Mixedwood stands can be less prone to severe wildfires compared to pine-dominated stands during the dormant season, because deciduous oaks or other hardwoods are interspersed between pine canopies, reducing the continuity of crown fuels and the density of ladder fuels [12, 16, 23, 24, 53, 79]. In lowland forest stands, hardwoods such as red maple and sweetgum are less tolerant of fire than many oak species, but the use of prescribed fires and wildfires are less frequent [16, 47, 77]. Overall, the continued extensive use of prescribed fire and wildfire suppression contributes to oak and pine regeneration and likely favors the persistence of oak-pine mixedwood forests, consistent with Fig 1.

Many of the dominant species in oak-pine and hardwood-pine mixedwoods are also considered to be relatively resistant to changes in climate, and are distributed across wide geographical and elevational ranges, can tolerate degraded, resource-limited environments, and some species tolerate extreme ranges in hydrologic conditions (e.g., pitch and shortleaf pines) [20, 72, 80]. A number of the dominant species in the mid-Atlantic region have already displayed increases in productivity and WUE$_e$, as a result of increased ambient $CO_2$ concentrations driving reduced transpiration [81, 82] and enhanced photosynthetic assimilation rates [43, 44]. In a previous study using LANDIS II to simulate future interactions of wildfire and climate in forests of the PNR, these factors were predicted to have only moderate effects on productivity of the major tree species, primarily because of their tolerance to drought stress and capacity to recover quickly from wildfires [34, 72].

Finally, our study provides some insight into the value of incorporating oak-pine mixed-woods into management strategies for forests in the mid-Atlantic region. Although mixed composition stands are typically more expensive to manage, they provide a greater variety of forest products when harvested selectively or thinned [20, 83]. As these forests age, simulating natural successional processes (e.g., forest thinning, mortality, regeneration) or delaying them through the use of prescribed fire and other silvicultural management practices would create more resistant forests [84–86]. Over time, treatments including prescribed burning, mechanical thinning, and selective cutting could reduce mortality of commercially important species while stimulating regeneration of key oak and pine species. By diversifying age class distributions and further enhancing forest heterogeneity, multi-aged mixedwoods management strategies may be particularly successful [22, 62, 87]. During and following infestations, lower levels of tree and sapling defoliation result in a more rapid recovery of leaf area and productivity, and reduced mortality decreases the amount of standing dead and coarse woody debris contributing to ecosystem respiration. Thus, one important benefit of mixedwood management is the faster recovery times of NEP to pre-infestation rates following insect infestations, maintaining forest carbon sequestration rates with only minor alterations to hydrologic resources. Forest insects have already shown us how effective such management strategies could be.

## Conclusions

Insect damage is now the dominant disturbance in forests of the mid-Atlantic region. Insect infestations that target dominant tree species are altering forest composition and structure, resulting in stands that consist of mixtures of pines, oaks, and other hardwoods. Despite difference in forest composition, FIA data, process-based forest productivity models, and carbon flux measurements indicate that oak-dominated, oak-pine mixedwood, and pine-dominated forests typically have similar NPP and NEP on annual time scales. Oak-pine mixedwood stands may be relatively resistant to future outbreaks of defoliators and bark beetles, reducing economic losses associated with tree mortality, and potentially mitigating the short-term impacts to ecosystem functioning resulting from insect damage, especially carbon sequestration. Management strategies that incorporate oak-pine mixedwood stands may increase the supply of undamaged forest products and provide better continuity in ecosystem services despite projected increases in forest insect infestations associated with changing climate.

## Supporting information

**S1 Table. Productivity of undisturbed oak-dominated, mixed oak-pine, and pine-dominated forests in the mid-Atlantic region.** Data are net primary production estimated from USFS Forest Inventory and Analysis data (FIA, [9]) and forest inventory plots in the Pinelands National Reserve (PNR, [13, 30]), simulated net primary production using PnET CN, a process-based forest productivity model [30, 31], WxBGC, a second process-based forest productivity model based on BiomeBGC [32], and LANDIS II, a plot-based model that simulates forest composition, succession, disturbance and other ecological processes linked to the CENTURY succession extension (ver. 3) [33]. Estimated net ecosystem productivity is derived from FIA data, simulated using WxBCG and LANDIS II, and calculated from carbon flux measurements in the PNR [13, 30].
(PDF)

**S2 Table. Structural characteristics of the canopy and understory in oak, mixed, and pine stands.** Data are presented for the beginning of the study in 2005 before infestation by gypsy moth, and at the end of the study in 2018. Values are means ± 1 SE. Significance levels were

tested using ANOVAs and Tukey's HSD tests, and values indicated with different superscripts among stands are significantly different.
(PDF)

**S3 Table. Structural characteristics of the canopy and understory in uninfested areas and areas infested by southern pine beetle.** Values are means ± 1 SE. Significance levels were tested using paired sample T-tests, and values indicated with different superscripts among areas are significantly different.
(PDF)

**S4 Table. Meteorological sensors and eddy covariance equipment used to measure turbulence, net ecosystem exchange of $CO_2$ (NEE) and evapotranspiration (Et) at the oak, mixed and pine stands.**
(PDF)

## Author Contributions

**Conceptualization:** Kenneth L. Clark, Carissa Aoki, Matthew Ayres, John Kabrick, Michael R. Gallagher.

**Data curation:** Kenneth L. Clark.

**Formal analysis:** Kenneth L. Clark, Carissa Aoki.

**Funding acquisition:** Kenneth L. Clark, Matthew Ayres.

**Investigation:** Kenneth L. Clark, Carissa Aoki, Michael R. Gallagher.

**Methodology:** John Kabrick.

**Project administration:** Kenneth L. Clark, Carissa Aoki, Matthew Ayres, John Kabrick, Michael R. Gallagher.

**Resources:** Michael R. Gallagher.

**Supervision:** Kenneth L. Clark, Carissa Aoki, Matthew Ayres.

**Visualization:** Kenneth L. Clark.

**Writing – original draft:** Kenneth L. Clark.

**Writing – review & editing:** Kenneth L. Clark, Matthew Ayres, John Kabrick, Michael R. Gallagher.

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
