## [Decision Letter · Decision Letter 0]

4 Jan 2022

PONE-D-21-34003Insect Infestations and the
Persistence and Functioning of Oak-Pine Mixedwood Forests in the Mid-Atlantic
Region, USA.PLOS ONE

Dear Dr. Clark,

Thank you for submitting your manuscript to PLOS ONE. After careful consideration, we
feel that it has merit but does not fully meet PLOS ONE’s publication criteria as it
currently stands. Therefore, we invite you to submit a revised version of the
manuscript that addresses the points raised during the review process.

Your paper addresses the very interesting question of how insect infestations may
affect forest composition, carbon dynamics, and hydrological cycling in northeastern
forest stands.  This is a particularly important question as damage from pests, such
as gypsy moths and pine beetles, seems to be increasing.  While this could be an
important contribution to our approach to managing forests, I agree with the
reviewers that the paper has a lot of promise but needs some additional
strengthening and clarification.  As Reviewer #2 points out, there is a lot of
information in the introduction but the purpose is not as clear here as in the
discussion.  There are some very good recommendations that they make to improve
clarity and flow in the introduction.  There are a number of places where the
descriptions of the different sites surveyed is confusing.  Please note some of the
issues highlighted by both reviewers throughout the methods and the results.  In
particular, it is difficult to disentangle the labels for the pine sites that are
infested or not by southern pine beetle and treated (managed?) or not.  Or is the
treatment status ignored for the purposes of your comparisons?  Reviewer #2 has
provided extensive line by line suggestions that should be carefully addressed.

The discussion repeats a lot of the results but could be strengthened by discussing
the broader context and complicating factors such as fire, climate change, and
herbivory and reducing the reiteration of the results.  Figure 1 is very useful as a
framework for the paper and it should be revisited more substantially in the
discussion.  For example, there is little discussion of differential impacts in
uplands and lowlands but rather an emphasis on pine versus oak. I agree with
Reviewer #2 that there should be more discussion of fire, especially since fuels
were measured (Line 277) for at least some sites. Does fire in these systems both
prescribed and wild complicate conclusions about the effects of the infestation and
resilience to disturbance?  It would be useful to discuss some of the implications
of these findings and how they might apply in other contexts. 

Both reviewers provide some suggestions to improve the tables and figures.  For
example, Reviewer #1 suggests more contrast is needed for Figure 6b and should more
closely resemble 6a. You might even want to similarly improve the contrast of Figure
5.  The figures are generally helpful but there is no description in the figure
legends what the arrows on the graphs are indicating.

This paper has a lot of promise and could be a good addition to the literature with
some improvements.

Please submit your revised manuscript by Feb 18 2022 11:59PM. If you will need more
time than this to complete your revisions, please reply to this message or contact
the journal office at plosone@plos.org. When
you're ready to submit your revision, log on to https://www.editorialmanager.com/pone/ and select the 'Submissions
Needing Revision' folder to locate your manuscript file.

Please include the following items when submitting your revised
manuscript:A rebuttal letter that responds to each point raised by the academic
editor and reviewer(s). You should upload this letter as a separate file
labeled 'Response to Reviewers'.A marked-up copy of your manuscript that highlights changes made to the
original version. You should upload this as a separate file labeled
'Revised Manuscript with Track Changes'.An unmarked version of your revised paper without tracked changes. You
should upload this as a separate file labeled 'Manuscript'.

If you would like to make changes to your financial disclosure, please include your
updated statement in your cover letter. Guidelines for resubmitting your figure
files are available below the reviewer comments at the end of this letter.

We look forward to receiving your revised manuscript.

Kind regards,

Karen Root, Ph.D.

Academic Editor

PLOS ONE

“Partial support for this project was provided by USDA Forest Service Forest Health
and Monitoring Program grants NE-EM-F-13-01 to KC and NE-EM-B-12-01 to MA and
AK.”

We note that you have provided additional within the Acknowledgements Section. Please
note that funding information should not appear in the Acknowledgments section or
other areas of your manuscript. We will only publish funding information present in
the Funding Statement section of the online submission form.

“Partial support for this project was provided by USDA Forest Service Forest Health
and Monitoring Program grants NE-EM-F-13-01 to KC and NE-EM-B-12-01 to MA and
AK.”

4. We noted in your submission details that a portion of your manuscript may have
been presented or published elsewhere. [DETAILS AS NEEDED] Please clarify whether
this [conference proceeding or publication] was peer-reviewed and formally
published. If this work was previously peer-reviewed and published, in the cover
letter please provide the reason that this work does not constitute dual publication
and should be included in the current manuscript.

6. Please include captions for your Supporting Information files at the end of your
manuscript, and update any in-text citations to match accordingly. Please see our
Supporting Information guidelines for more information: http://journals.plos.org/plosone/s/supporting-information

Reviewers' comments:

Reviewer's Responses to Questions

**Comments to the Author**

1. Is the manuscript technically sound, and do the data support the conclusions?

Reviewer #1: Yes

Reviewer #2: Yes

2. Has the statistical analysis been performed
appropriately and rigorously? 

Reviewer #1: Yes

Reviewer #2: Yes

3. Have the authors made all data underlying the
findings in their manuscript fully available?

Reviewer #1: Yes

Reviewer #2: Yes

4. Is the manuscript presented in an intelligible
fashion and written in standard English?

Reviewer #1: Yes

Reviewer #2: Yes

5. Review Comments to the Author

Reviewer #1: Great work, just a few minor edits found.

Line 42 would be good to note that it is a decrease in sever wildfires.

Line 59-60 reword, for example: These "mixedwoods" are characterized by neither
hardwood nor softwood exceeding 75% dominance.

Line 87 says oak pine mixedwoods, but isn't it also making lowland deciduous pine
mixedwoods?

Line 177 you put a ; in 36,654 ha

Line 200-201 would be good to clarify timing by putting year(s) in parenthesis after
pre-, during, post-

Line 221 why was it 10-16? were some removed due to location issues?

Line 222 has "(see below)" but doesn't refer to anything, is there supposed to be a
Figure?

Tables with sub-sections should have that sub-section header in Bold or Italics to
help with reading the table.

Figure 6b would look better with more contrasting colors.

Reviewer #2: Overall, this paper contains a lot of information and data and gets a
bit confusing. The authors do not adequately set up the paper in a way that allows
the reader to follow the results. However, I think with some tweaking, it could be a
really great paper.

6. PLOS authors have the option to publish the peer
review history of their article (what does this mean?). If published, this will
include your full peer review and any attached files.

If you choose “no”, your identity will remain anonymous but your review may still be
made public.

**Do you want your identity to be public for this peer review?** For
information about this choice, including consent withdrawal, please see our
Privacy Policy.

Reviewer #1: No

Reviewer #2: No

---

## [Author Response · Author response to Decision Letter 0]

28 Feb 2022

Detailed replies to comments by the Editor: 

Your paper addresses the very interesting question of how insect infestations may
affect forest composition, carbon dynamics, and hydrological cycling in northeastern
forest stands. This is a particularly important question as damage from pests, such
as gypsy moths and pine beetles, seems to be increasing. 

While this could be an important contribution to our approach to managing forests, I
agree with the reviewers that the paper has a lot of promise but needs some
additional strengthening and clarification. As Reviewer #2 points out, there is a
lot of information in the Introduction, but the purpose is not as clear here as in
the discussion. There are some very good recommendations that they make to improve
clarity and flow in the Introduction. 

Thank you for the encouraging set of comments. We agree with the Editor’s and
Reviewer #2’s assessments of the Introduction section, and have reorganized some of
these paragraphs in the revised manuscript. Two paragraphs that were not essential
to developing the research questions and objectives have been edited; and 1) the
paragraph that discussed “associative resistance” has been removed and integrated
with the Discussion section, and 2) the paragraph on the productivity (NPP and NEP)
of undisturbed oak-dominated, mixedwood, and pine-dominated stands has been
condensed, with the extensive Table 1 moved to Supplemental S1 Table. 

There are a number of places where the descriptions of the different sites surveyed
is confusing. Please note some of the issues highlighted by both reviewers
throughout the methods and the results. 

We have condensed and moved the sub-section on “L. dispar and southern pine beetle”
in the Materials and Methods section. We had intended this to summarize the impacts
and spatial distributions of recent infestations in the region, but it was probably
more confusing that helpful. We now only describe the spatial extent of the two
recent infestations that we studied in Pinelands National Reserve of New Jersey in
the appropriate subsection for forest census measurements. 

Further, we have reversed the order of presentation of the FIA-type plots sampled
pre-, during and post L. dispar infestations for clarity. We have also rewritten the
description of the FIA-type plots that were sampled for southern pine beetle
infested areas, highlighting that only untreated stands were used in our analyses
here (please see comments below). 

In particular, it is difficult to disentangle the labels for the pine sites that are
infested or not by southern pine beetle and treated (managed?) or not. Or is the
treatment status ignored for the purposes of your comparisons? 

We first must apologize for a typographic error on the label for pine sites in Figure
4, which from left to right presents the relative basal areas of an oak-dominated
stand pre- and post-infestation of L. dispar, a mixed stand at the beginning of the
study, and pine-dominated stands post- and then pre-infestation of southern pine
beetle, with the changes driven by infestations indicated with arrows. We had
inadvertently switched the order of “post-“ and “pre-“ for the pine sites, and have
corrected this in the revised version. 

Although we measured forest structure in stands that were untreated and had been
treated for southern pine beetle infestations (treatments included “cut and leave”
and “cut and chip”; detailed in an annual report for the USFS Forest Health and
Monitoring program in Clark et al. 2017 [24]), we have only used data from the 10
untreated, naturally occurring infestations here. We omitted data from the sites
where suppression treatments were conducted because a major difference between
untreated and treated stands is that pine saplings were cut or damaged in the
treated stands, and this obscured the impacts of southern pine beetle and the shift
to stands that more closely resemble mixedwood composition, as shown in Figure 4. We
have rewritten our description of this in the Methods section, and now mention the
treatment types explicitly in the Methods section. 

Reviewer #2 has provided extensive line by line suggestions that should be carefully
addressed.

Reviewer #2 has provided very helpful line by line comments, and we have attempted to
address all of these. We feel following these suggestions has improved the clarity
of the manuscript, and we appreciate the time Reviewer #2 spent on reviewing our
previous draft. 

The discussion repeats a lot of the results but could be strengthened by discussing
the broader context and complicating factors such as fire, climate change, and
herbivory and reducing the reiteration of the results. 

Both the Editor and Reviewer #2 suggested that the Discussion section could be
improved by first reducing the reiteration of the Results, and then expanding the
broader context and complicating factors. In the original version of the manuscript,
we intended to summarize the Results in the Discussion section first, noting that we
are presenting a complex set of results that has integrated data from long-term
forest census plots, FIA-type sampling, and long-term flux data from three sites
that have been variously disturbed by insect infestations and prescribed fires.
However, we agree with the Editor and Reviewer #2 and have removed the sub-sections
headings and condensed the three sub-sections on impacts of L. dispar and southern
pine beetle on forest composition, structure and productivity into three paragraphs.
Further, we have expanded linkages to the conceptual model in Figure 1 by
referencing this where appropriate, as per the comment below. 

Figure 1 is very useful as a framework for the paper and it should be revisited more
substantially in the discussion. For example, there is little discussion of
differential impacts in uplands and lowlands but rather an emphasis on pine versus
oak. 

In the original version of the manuscript, we largely limited our discussion to
upland systems, because all of our research on L. dispar and the three carbon flux
towers are located in upland forest stands. While we believe that a more extensive
treatment of lowland forests would be interesting, we felt this would be beyond the
scope of our analyses. However, we have expanded our references to southern pine
beetle effects in lowland systems throughout the revised manuscript. We also have
expanded our discussion of fire return intervals and the fact that hardwood tree
species in lowland forests, primarily red maple and black gum, are more fire
intolerant than oaks in the revised Discussion section. 

I agree with Reviewer #2 that there should be more discussion of fire, especially
since fuels were measured (Line 277) for at least some sites. Does fire in these
systems both prescribed and wild complicate conclusions about the effects of the
infestation and resilience to disturbance? 

We have expanded our discussion of fire, which does appear throughout the original
version of the manuscript but was not highlighted particularly well. 

Your question is an excellent one, and throughout the Discussion of the revised
manuscript we have attempted to show that fire, especially the current patterns of
extensive use of prescribed fire and wildfire suppression, would tend to reinforce
the persistence of uneven age mixedwood stands because it promotes the regeneration
of both oaks and pines by reducing understory competition and removing excess litter
layer on the forest floor. An abundance of research has been conducted on the
effects of low intensity fire in the Pinelands National Reserve and through the
mid-Atlantic region that we now cite in the revised manuscript. In addition, we now
explicitly cite how the effects of insect infestations and fire are consistent with
a recently published conceptual model of mixedwood formation and persistence (Kern
et al. 2021 [22]}. 

It would be useful to discuss some of the implications of these findings and how they
might apply in other contexts. 

We have attempted to expand our discussion of the implications of our study
throughout the revised Discussion section. 

Both reviewers provide some suggestions to improve the tables and figures. For
example, Reviewer #1 suggests more contrast is needed for Figure 6b and should more
closely resemble 6a. You might even want to similarly improve the contrast of Figure
5. The figures are generally helpful but there is no description in the figure
legends what the arrows on the graphs are indicating.

We have used better contrasting colors for Figures 5a and 5b, and 6a and 6b. We have
also provided a description of what we intended the arrows to indicate in these two
figures. The sentence stating, “Arrows indicate the directional changes in forest
structure and composition following L. dispar infestations.” has been added to the
legend below Figures 5 and 6. 

This paper has a lot of promise and could be a good addition to the literature with
some improvements.

Thank you again for supporting our manuscript 

Detailed replies to comments by Reviewer #1: 

Reviewer #1: Great work, just a few minor edits found.

Thank you.

Line 42 would be good to note that it is a decrease in sever wildfires. We have added
the phrase “…and a decrease in the occurrence of severe wildfires [1-3]. 

Line 59-60 reword, for example: These "mixedwoods" are characterized by neither
hardwood nor softwood exceeding 75% dominance. We have reworded this sentence to
read, “These “mixedwoods” are characterized by neither hardwoods or softwoods
exceeding approximately 75% dominance [e.g., 19-21].”

Line 87 says oak pine mixedwoods, but isn't it also making lowland deciduous pine
mixedwoods?

We agree. We believe that Reviewer #1 intended to mean hardwood pine mixedwoods, and
so we have added a phrase to mention this. We have also pointed this out explicitly
throughout the Discussion section.

Line 177 you put a ; in 36,654 ha. The semi-colon is now a comma in 36,654 ha for the
acreage of wildfires from 2004 to 2016. 

Line 200-201 would be good to clarify timing by putting year(s) in parenthesis after
pre-, during, post-. Thank you for pointing this out. We have added the years in
parenthesis for each of these periods. 

Line 221 why was it 10-16? were some removed due to location issues?

Yes, some plots fell on paved or sand roads, or in the case of the pine dominated
site an unforested fire break. These were either not sampled or omitted from our
analyses here. We now state this more clearly in the revised manuscript. 

Line 222 has "(see below)" but doesn't refer to anything, is there supposed to be a
Figure?

Tables with sub-sections should have that sub-section header in Bold or Italics to
help with reading the table.

This statement referred to the flux towers, but we agree this was confusing. We now
state “(described below)” to clarify. 

Thanks, this is a helpful comment for table presentation. We have reformatted all of
the tables with sub-section headers in bold. 

Figure 6b would look better with more contrasting colors.

We have increased the contrast in Figures 5 and 6 by lightening the color of the bars
indicating water use efficiency values for spring periods. 

Detailed replies to comments by Reviewer #2: 

This paper looks at successional changes in forests due to outbreaks of SPB and L.
dispar. 

It appears as though there are a couple different objectives, but those are never
clearly stated in the manuscript, so it doesn’t really become clear what the paper
is about until the discussion. 

There is a lot going on in this paper and, at times, can get confusing to read. The
objective in the abstract is to “understand ecological consequences of invasive
insects on…..” but that’s pretty vague. 

Also, you discuss southern pine beetle which is not an invasive insect under
some/many definitions. 

Overall, the paper leaves out some major details in terms of objectives and methods.
I believe the authors did a massive amount of work on this and simply need to be
more specific and intentional in their writing. My other main issue is with
consistency in writing and explanations.

These are all very helpful comments. We do agree that this is a complex paper, and
have reorganized and rewritten the Introduction section so that our objectives are
easier for readers to follow. Following this, we have condensed some of the
subheading topics throughout the Methods and Discussion section, for example the “L.
dispar and southern pine beetle” section has been condensed into the sections on
forest census measurements for L. dispar and southern pine beetle. We have also
rewritten much of the Discussion section. 

General comments:

Change all instances of “gypsy moth” to Lymantria dispar as the common name is being
changed. We have replaced “gypsy moth” with “Lymantria dispar” at first use and “L.
dispar” throughout the remainder of the manuscript. 

The authors define phrases/words many times throughout the paper while still
continuing to spell them out. At the same time, some things are stated but never
defined. For instance, ecosystem water use efficiency is define at least 3 times
while the authors don’t use the acronym WUE. 

We apologize for the inconsistencies. We have defined all terms at first use, then
use the correct acronyms throughout the revised manuscript. 

Introduction:

The intro feels a bit out of order. 

We agree, and as noted above, we have rewritten the Introduction section so that
relevant material is covered in such detail, and the development of our questions
and then objectives are clearer. 

L43: Add commas around “…and intensity” Thank you, this is clearer now.

Remove L52-54. 

We have substituted this introductory sentence, and now introduce the Pinelands
National Reserve here, as Fig 1 (the conceptual model) addresses forests in the PNR. 

L70-71: This sentence implies that the authors are going to discuss vulnerable
species somewhere but this doesn’t come up throughout the paper. 

We agree with Reviewer #2 that the use of the term “vulnerable” is vague and could be
interpreted as meaning the conservation status of a species. Thus, here and on line
509 we have omitted the term “vulnerable” and have reworded to “susceptibility to
insect infestations…”

L92: add “primary” to NPP definition OK, now corrected. 

L114: This whole paragraph could be much earlier, I think. 

We agree because this is a key objective and differs from the Abstract. We have
rewritten much of the Introduction of the revised manuscript to address this and
comments above. 

L122: The authors have not defined NEE yet. We now define “net ecosystem exchange of
CO2” before the first use of “NEE”

L137: Define and replace “course woody debris” with CWD 

We now define course woody debris as CWD on first use and use CWD throughout the
remainder of the revised manuscript. We also report values for the oak, mixed and
pine stands, and refer to previously published values. Thanks, this was not clear in
the last version. 

- 

L146: Remove “…as summarized in the conceptual model in Fig 1” and imply cite (Fig 1)
at the end of the sentence. We have shortened this sentence as suggested, thanks. 

L147-148: Sounds like this is what the authors are setting up to investigate but this
is different than the “objective” in the abstract. 

We have reworded the single objective listed in the Abstract so that it is a better
description of actual objectives of our study. We have also revised much of the
Introduction section. 

L153-155: Define these as acronyms and then consistently use the acronyms
throughout.

We have now defined all terms and acronyms at first use and used the correct acronyms
throughout the manuscript. Again, we apologize for the inconsistencies. 

L245: the authors use N for nitrogen here but in other places it is spelled out. Be
consistent.

We have used the abbreviation “N” for nitrogen throughout the manuscript, except at
first use where it is defined, and when it appears at the beginning of a sentence.
Thank you, this is clearer now. 

Methods:

There are SDs for precip but not temp. Don’t think SD are needed at all in site
descriptions. 

In the revised manuscript, we have followed a standard protocol on reporting means
and SD’s for air temperature and precipitation, reporting averages over the last 30
years. We have added SD values to the temperature data presented in the text. 

The sections on L. dispar and SPB all seems like intro material.

We considered moving these two paragraphs to the Introduction, but believe that they
are best shortened and included as part of the Materials and methods section.
However, we have removed the general information on susceptible species and only
reported the years and extents of these infestations in southern New Jersey, under
the descriptions of the forest census plots. This is important information to
include somewhere, because it does indicate the extent to which L. dispar and SPB
have impacted forests in the Pinelands National Reserve. 

L222: If plots were set up in a 4 x 4 arrangement, how could there be 10-16 plots?
Should always be 16….

This is true, but some plots were not sampled or not included in the analyses because
one or more of the FIA subplots fell on sand roads or disturbed, non-forested areas.
We apologize, this was not written very clearly in the previous version. 

L226: Capitalize DBH and use throughout. Don’t need 1.27 m as DBH already has a
definition. 

We have changed all uses of “dbh” to “DBH”. 

Height should have (m) after it to show how you measured. How was crown condition
assigned? There is no description of crown condition anywhere. 

We used the standard Forest Inventory and Analysis protocol for assigning crown
classes for trees. These are emergent, dominant, co-dominant and suppressed. We do
not present those data here, but would include it in the archived data. 

L230: Why is recruitment in here twice? We apologize for this typographical error.
This sentence now reads correctly. 

L231: What is a clip plot?

These were destructively harvested plots measuring either 1.0 m2 or 0.5 m2 used to
determine the aboveground biomass of understory vegetation and saplings. We have
reworded this sentence in the revised manuscript. 

L251-252: Did you extract data from these papers and then add them to your
analyses?

These cited papers had additional [N] data for growing season foliage of the dominant
and co-dominant species that we sampled. We compared our results to theirs, and
their averaged values were reasonably close to ours. 

L254: use [N] instead of spelling out “concentration” each time. Also, is content
different than concentration? Again, these things are not well defined.

We now define N concentration as [N] following first use. We also define N content
clearly, and use this term throughout the manuscript. Thank you for pointing this
out, it is clearer now. 

L267: Why use SD instead of SE? Also, this seems like a huge SD!

We used the SD value because that was reported by Aoki et al. in the cited
publication. 

These values came from tree ring counts of cored trees in stands sampled by Aoki et
al., and their sampling occurred in the same stands we report here (their transects
were co-located with the FIA type plots we installed and sampled in some stands). 

L274: How was cover measured? This is not clear.

Cover was visually estimated from 4 cardinal directions out from the center point of
each plot and then averaged for cover of understory vegetation and tree saplings. We
have added “visually estimated“ in the text description. We would also include these
values in the archived data. 

L277: I’m confused as to what fuels have to do with anything. Fuels (like for fires?)
have not been brought up at all yet.

We do mention wildfires and prescribed fires early in the Introduction, and effects
of fire on species composition are also mentioned when describing the conceptual
model in Figure 1 in the revised manuscript. We then return to the effects of fire
in the Discussion, where we discuss the importance of fire in the regeneration of
pines and oaks, and also describe how mixedwood forests may be less prone to severe
wildfires compared to pine-dominated forests which have greater amounts of ladder
and crown fuels. We have removed the mention of “available fuels” for pine trees and
saplings here because we do not report these values in Table S2, which presents
structural characteristics of areas infested by southern pine beetle and uninfested
areas. Again, we would report these in the archived datasets. 

L285: NEE should be defined earlier and should be the strict definition (i.e., net
ecosystem exchange). We have defined NEE at first use in the revised manuscript. 

L315: How did you assume it was dry? Were there certain environmental variables you
checked beforehand? If so, then it’s not really an assumption per se. 

We couldn’t really measure amounts of the water on leaf, needle and other canopy
surfaces, so we thought it more accurate to use the term “assumption” here. We
followed the protocol in previously published accounts of calculations for water use
efficiency, WUEe., and used local half-hourly precipitation data to estimate dry
periods. 

- 

L321: First time basal area is mentioned. Should be defined as BA with BA being used
throughout the rest of the paper.

We have now defined “basal area” as BA at the first use, and now use the term BA
throughout the remainder of the manuscript. 

L332-334: This should be the first thing in the stats section

This is a good point. We have moved this sentence from the end of the paragraph to
the beginning, because this was tested first before proceeding with the statistical
analyses. 

L323: Software used should be at the end (assuming you used that software for all
analyses)

We did use SYSTAT 12 for all of the statistical analyses, and have moved this
sentence to the end of the paragraph in the Statistics section. 

L330: How many subsets? We have reorganized this sentence for clarity, and now report
this value as “25 subsets”. 

Results

L339-347: Results are very vague (i.e., “….and CWD were low at all three stands…”).
What is “low”?

We have chosen to present the results for L. dispar chronologically, following their
impact on 

the three stand types because this highlights their differential impact on forest
composition and structure. We now present CWD values throughout the Results section
in the revised manuscript. Thank you, this was too vague in the previous version. 

L372: “Stem increment”? We have substituted the term “BA” for “stem” to be consistent
with the use of basal area, and for clarity. 

- 

L378-379: Are these values ± SD or SE? We have now defined these values above as
SE.

- 

L382: Again, results are vague with “very low” values and no means or other
descriptors that let us know what “very low” actually means. We agree that this is
vague and have added values to the text for coarse wood. 

L400-407: These types of results are not given for L. dispar. This is more what I
expect in a results section. 

- 

L426: What is “course wood mass”? Do you mean CWD? Yes, and we have changed this to
CWD throughout the manuscript. This seems clearer now. 

L430-435: Seems like an intro sentence to the discussion.

Discussion

L505-507: Can you generalize this to “insect infestations”? You only looked at one
herbivore and one bark beetle…

This is true, and we have rewritten this sentence in the revised manuscript to
indicate that we only investigated the effects of the two forest insects. 

L509: This is the first time that “vulnerable” has come up again. By vulnerable, do
you mean listed or simply susceptible to herbivory? Not really sure what vulnerable
means in the context of this paper. We agree, and have omitted the term “vulnerable”
throughout the text, because what we intend is “susceptible to herbivory” 

Were there any areas that had BOTH SPB and L. dispar?

This is a good question. We observed stands with patches of oaks which likely had
been previously impacted by L. dispar (as indicated by larger dead and damaged oak
trees) adjacent to SPB stands that had been treated using cut and leave or cut and
chip treatments. We sampled the SBP portions of these stands, but unfortunately
because treatments had been conducted in the SBP portions of the stands, we did not
use that information here because treatments also reduced the basal area and biomass
of pine saplings. Perhaps in the next infestations? 

L519: First time fire management has been mentioned. Is this related to “fuels” that
came up earlier? Or unrelated? Fire really isn’t mentioned elsewhere.

We have attempted to highlight fire throughout the revised manuscript. We have also
strengthened the discussion about use of prescribed fires and wildfires in
regenerating pine and oaks, and fire-intolerance in some hardwoods. Thank you, this
was a very good comment, and we attempted to improve the discussion of fire
throughout the revised manuscript. 

L533: Are you using N content as a proxy for “foliage quality”? If so, this isn’t
defined.

Yes, and we now define this in paratheses in this sentence. Thanks for pointing this
out, because it makes our intended use of the N content information clearer. 

L599: Add end parentheses to the end of the sentence. OK, thank you. 

L602-605: Wildfires are brought up here but, again, it’s not clear whether this was
something actually looked at by the authors. 

We now provide a number of citations throughout the manuscript that have either
investigated the reduced occurrence of wildfires because of suppression activities
or simulated the impacts of wildfires on forest composition, structure and ecosystem
functioning. We also mention how the pine dominated stand had been burned in a
wildfire in 1995, and subsequently in prescribed fires in 2008 and 2013, and that
three prescribed fires have been conducted at the mixed pine-oak stand. Further, we
have rewritten a paragraph in the Discussion section summarizing the documented the
effects of prescribed fire in promoting the regeneration of oaks and pines in the
PNR and throughout the mid-Atlantic region. This is important information for
understanding the persistence of uneven age mixedwood stands that we did not treat
very effectively in the previous version of the manuscript, and we appreciate this
set of comments. 

Tables and Figures:

Figure 1: Shouldn’t it be “southern pine beetle infestations”? You didn’t look at any
other “pine beetles”. 

Yes, this is true. We have substituted the term “Southern pine beetle infestations”
in Figure 1. We have also changed “Gypsy moth” to “Lymantria dispar” for
consistency. 

Table 1: Add “(PNR)” to caption after you define Pinelands National Reserve as the
authors use PNR in the table but do not define it. We have added “(PNR)” to the end
of the last sentence of the caption for Table 1. 

Table 2: I don’t know that a column with “figure” is necessary here.

We have left the references to the figures in the Tables because we feel that it will
allow readers to find significance levels for statistical tests easily. 

Figure 2 caption: This should stand alone. The authors don’t define “Inf” here.

We now define Pre, Inf, and Post in the caption for Figure 2, using the following
definition: 

“Pre” indicates before infestations, “Inf” indicates during infestation in 2007, and
“Post” indicates a decade following infestations. 

Table 3: Again, I don’t think a column for figure is necessary. And, since the
authors simply report whether p-values are sig or not, this could be done with an *
next to the T value instead and take up considerably less space. Table 5: Same
comments as above.

Following the comment above, we left the references to the figures in each table. We
feel that this will allow readers to find this information more easily. 

Check references. Some are abbreviated while others are not. Some journals are
abbreviated while others are not. Make sure you are consistent with journal
requirements.

We have reformatted the references so that they are consistent with PLoS One
instructions. We apologize for this error. 

Editorial comments for PLoS One formatting 

For Point #1 and the last comment by Reviewer #2, we have paid much closer attention
to the proper formatting for PLoS One manuscripts, including the Tables and
References. 

We now provide information on how we obtained permission to sample sites for both
sets of forest census plots. Nearly all lands were New Jersey state forests or
wildlife management areas managed by the New Jersey Department of Environmental
Protection. 

3. Please remove any funding-related text from the manuscript and let us know how you
would like to update your Funding Statement. 

For Point #3, we have removed the funding-related text from the Acknowledgement
section in the revised manuscript. We would like our online funding statement to
remain unchanged, and read: “Partial support for this project was provided by USDA
Forest Service Forest Health and Monitoring Program grants NE-EM-F-13-01 to KC and
NE-EM-B-12-01 to MA and AK.”

Thank you for changing this on the online submission form. 

For Point #4, we would first like to clarify that some of the information in the
manuscript has been previously published, and where we have used previously
published information we have provided the appropriate citations. In summary, forest
census data and carbon flux data from the three sites used to analyze the impacts of
L. dispar previous to 2016 have been published in peer-reviewed publications. Forest
census and carbon and hydrologic flux data for the “post” period in 2018 are unique
to this manuscript. Similarly, summaries of the forest census data from the stands
infested by southern pine beetle have been published in an annual report for the
USFS Forest Health program, and in a meeting proceedings in 2020. These are also
cited where used in this manuscript. Using this information is essential in setting
the documenting the pre-infestation conditions for L. dispar and southern pine
beetle, and for documenting some of their impacts through time. The overall
conceptual model of mixedwood forest formation and persistence, and the analyses of
forest productivity and hydrologic data in the context of mixedwoods is unique to
this manuscript.

to Reviewers.docx
---

## [Editor Report · Decision Letter 1]

11 Mar 2022

Insect infestations and the persistence and functioning of oak-pine mixedwood forests
in the Mid-Atlantic Region, USA.

PONE-D-21-34003R1

Dear Dr. Clark,

We’re pleased to inform you that your manuscript has been judged scientifically
suitable for publication and will be formally accepted for publication once it meets
all outstanding technical requirements.

Kind regards,

Karen Root, Ph.D.

Academic Editor

PLOS ONE

Additional Editor Comments (optional):

I appreciate the authors’ thoroughness and thoughtfulness in addressing the numerous
comments and suggestions by the reviewers. The revisions have substantially improved
the clarity and increased the flow while strengthening the main conclusions of the
paper. With these revisions the paper is now suitable for publication and
significantly advances our understanding of the complex interactions of insects and
forests.
---

## [Editor Report · Acceptance letter]

25 Apr 2022

PONE-D-21-34003R1 

Insect infestations and the persistence and functioning of oak-pine mixedwood forests
in the Mid-Atlantic region, USA. 

Dear Dr. Clark:

I'm pleased to inform you that your manuscript has been deemed suitable for
publication in PLOS ONE. Congratulations! Your manuscript is now with our production
department. 

Kind regards, 

on behalf of

Professor Karen Root 

Academic Editor

PLOS ONE